



# In situ observations of stratospheric HCl using three-mirror integrated cavity output spectroscopy

Jordan Wilkerson[1], David S. Sayres[2], Jessica B. Smith[3], Norton Allen[2], Marco Rivero[2], Mike Greenberg[2], Terry Martin[2], and James G. Anderson[1,2,3]

[1]Department of Chemistry and Chemical Biology, Harvard University, Cambridge, MA 02138, USA
[2]Paulson School of Engineering and Applied Sciences, Harvard University, Cambridge, MA 02138, USA
[3]Department of Earth and Planetary Sciences, Harvard University, 12 Oxford Street, Cambridge, MA 02138, USA

*Correspondence to*: Jordan Wilkerson (jwilkerson@g.harvard.edu)

**Abstract.** Stratospheric HCl observations are an important diagnostic for the evaluation of catalytic processes that impact the ozone layer. We report here in situ balloon-borne observations of HCl employing an off-axis integrated cavity output spectrometer (ICOS) fitted with a re-injection mirror. The spectrometer has a 90 % response time of 10 s to changes in HCl and a 30 s precision of 26 pptv. The instrument was deployed alongside an ozone instrument in August 2018 on a balloon-borne descent between 20-80 hPa (29-18 km altitude). The observations agreed with nearby satellite measurements (MLS) within 10 % on average. This is the first time that stratospheric measurements of HCl have been made with ICOS and the first time any cavity enhanced HCl instrument has been tested in-flight.

## 1 Introduction

Most of the hydrogen chloride (HCl) in the stratosphere exists because of the cumulative emissions of chlorofluorocarbons (CFCs), whose widespread use as non-toxic refrigerants began in the 1950s (WMO 2018). CFCs have been largely phased out globally since the Montreal Protocol in 1987 and subsequent London and Copenhagen amendments because their degradation via photolysis in the stratosphere leads to the catalytic destruction of the ozone layer, a layer that protects life from the Sun's destructive ultraviolet radiation (Molina & Rowland 1974). Though some emissions of CFC-11 are still being detected in the Northern Hemisphere, atmospheric CFC levels have been declining for decades (Montzka et al., 2018). Despite this, elevated chlorine levels persist in the stratosphere (Mahieu et al., 2014). In 2016, approximately 80 % of the chlorine input from the troposphere to the stratosphere still came from substances controlled under the Montreal Protocol (WMO 2018).

Stratospheric HCl is considered an inorganic chlorine reservoir because the bonded hydrogen renders the chlorine atom unreactive to ozone. When environmental conditions are favorable, however, HCl can react, liberating chlorine from its reservoir into a catalytically active form. Three stratospheric perturbations can increase the rate at which HCl is converted to reactive chlorine species on the surface of liquid or solid aerosols: (1) lower temperatures, (2) enhanced water vapor levels, and (3) enhanced aerosol levels (Anderson et al., 2012; Solomon, 1999). Deep convective storm systems have been observed over the Great Plains of the United States that can enhance stratospheric water vapor levels (Hanisco et al., 2007;



Smith et al., 2017). These water vapor enhancements can persist in the lower stratosphere over the U.S. because they are trapped in the North American Monsoon Anticyclone for periods of a few days to over a week (Weinstock et al., 2007; Clapp et al., 2019). Alternatively, sulfate aerosol levels can be enhanced by volcanic eruptions or by solar geoengineering strategies, which propose to inject sulfate particles into the stratosphere as a way to mitigate climate change by reflecting solar radiation
back to space (Solomon et al., 1999; Keith, 2000; Tilmes et al., 2009). These environmental changes all increase the probability of transient ozone depletion over the U.S. and other parts of the world.

With increased forcing of the climate by increasing greenhouse gas (GHG) levels, severe summer storms that lead to deep convective injection in the U.S. may increase in number or intensity, while stratospheric temperatures are expected to decrease (Diffenbaugh et al., 2013). Furthermore, as climate change gets more severe, the consideration for solar geoengineering
strategies to complement GHG emission reductions will likely be taken more seriously; the need for research into the effects of sulfate aerosol injection on stratospheric chemistry will become correspondingly greater. Therefore, an established strategy for monitoring the stratosphere is important for understanding the extent to which the expected stratospheric perturbations over the next century translate to perturbations in its chemical composition.

HCl monitoring is an essential component of a platform that seeks to evaluate the effects of stratospheric perturbations. HCl
is quick to respond to perturbations in temperature, water vapor, and aerosol levels (for details into these reaction mechanisms, see Anderson et al., 2012 and Anderson et al., 2017). The heterogeneous catalytic conversion from HCl to the radical chlorine monoxide (ClO) may occur only where there is enhanced water vapor from convective injection. Therefore, observations of HCl with high spatial resolution are of particular importance. In situ HCl measurements are also of interest because the molecule can serve as a tracer for stratospheric air masses and ozone in the lower stratosphere (Marcy et al., 2004), and it can
provide better understanding of the total stratospheric chlorine budget (Bonne et al., 2000). Satellite monitoring systems do currently exist and provide invaluable information about the stratosphere's chemical composition, including HCl. The main example is the Earth Observing System (EOS) Microwave Limb Sounder (MLS), which is located on NASA's Aura satellite (Waters et al., 2006). MLS, however, has limited spatial and temporal resolution and a set trajectory that limits the number of local environmental phenomena that can be observed. In situ, airborne measurements that can attain higher horizontal and
vertical resolution are essential for understanding how perturbations of the future stratosphere may alter the delicate balance established by the chemicals that compose it. Previous in situ measurements of stratospheric HCl have used either multi-pass cells (ALIAS-I, Webster et al., 1994; ALIAS-II, Scott et al., 1999 and Christensen et al., 2007) or chemical ionization mass spectrometry (CIMS, Marcy et al., 2005). While measurements made via the CIMS technique have excellent precision, the technique has greater mass and volume requirements compared to optical approaches, and it requires constant in-flight
calibration (Roberts et al., 2010).

Recently, several HCl instruments based on cavity enhanced spectroscopy have been developed which have the advantage of longer effective path lengths (>1 km) in a small cell compared to multi-pass spectrometers, though none have been flown. Two of these are commercial spectrometers introduced within the past decade by Tiger Optics (Tiger, 2020) and Los Gatos Research (LGR, 2020), both of which use off-axis integrated cavity output spectroscopy (OA-ICOS). ICOS is a cavity-enhanced



absorption technique that has been used to detect key trace gases both in the troposphere and in the stratosphere (e.g. Sayres et al., 2009; Dyroff et al., 2014; Wilkerson et al., 2019). However, these commercial HCl instruments are designed more for ground-based industrial applications, and their response times are relatively slow (around 1 minute). The only cavity-enhanced HCl spectrometer reported in peer-reviewed scientific literature relies on cavity-ringdown spectroscopy (CRDS) rather than ICOS to realize the desired precision (Hagen et al., 2014). The 90 % response time of this instrument is 10 seconds,

with a 1 $\sigma_{60s}$ precision of 20 parts per trillion by volume (pptv). It was only tested in comparison with ground-based observations. Airborne campaigns introduce technical challenges not faced by ground-based campaigns. For example, airborne campaigns entail large variations in temperature and humidity as well as turbulence that can compromise the mode matching stability between the laser and cavity that CRDS often depends on for enhanced signal (Ouyang et al., 2012).

We report here a flight-tested OA-ICOS instrument for measuring HCl that can contribute to a future airborne platform for in

situ stratospheric monitoring. The instrument was fitted with a third mirror, called a re-injection mirror (RIM), which amplifies the light intensity within the optical cavity, thus increasing the amount of light delivered to the detector (Leen et al., 2014). This HCl instrument was deployed along with an ozone instrument on a balloon-borne NASA campaign, called the Harvard University Stratospheric Chemistry Experiment (HUSCE). The instrument platform was launched from the Columbia Scientific Balloon Facility at Fort Sumner, New Mexico on August 24, 2018. This is the first reported in-flight demonstration

1) of a RIM-ICOS instrument, and 2) of any cavity-enhanced instrument that measures HCl. We discuss here the details of this instrument, its performance in the laboratory, and comparison of the August 2018 test flight results with other stratospheric measurements of HCl.

## 2 ICOS HCl instrument description

### 2.1 Spectroscopic overview

The molecular absorption line that the RIM-ICOS instrument engages to measure HCl is at 2963 cm$^{-1}$ (3.37 μm) (Li et al. 2011). This absorption line is part of HCl's fundamental ro-vibrational transition (Figure 1). Previous measurements of stratospheric HCl observed nearby transitions at 2926 cm$^{-1}$ (e.g. Ackerman et al., 1976; Farmer et al., 1976; Williams et al., 1976), at 2945 cm$^{-1}$ (Webster et al., 1994), and at 2799 cm$^{-1}$ (Christensen et al., 2007).

To be a useful absorption feature for measurement, the HCl transition must be spectroscopically isolated from transitions by

other molecules that occur in the stratosphere. Transitions from other molecules can obfuscate the signal, limiting both the accuracy and precision of the HCl measurement. When selecting the line, the spectral interference from several common stratospheric molecules was considered, including: $CH_4$, $CO$, $CO_2$, $H_2O$, $NO$, and $N_2O$. A theoretical spectrum of the typical stratosphere is shown in Figure 2 as the black line. The only significant transitions in the vicinity of the absorption feature we observe are due to $H_2O$. The absorption feature used to measure HCl has no interference by either water vapor or other

molecules in the stratosphere. Because the campaign was balloon-borne, elevated water vapor was observed sporadically due



to outgassing from the balloon (up to ~150 ppmv $H_2O$). Even under the most extreme instances of balloon interference, only a modest absorption feature by $H_2^{18}O$ is present (Figure 2, blue line).

## 2.2 HCl instrument layout

The photon source used for the HCl instrument is a 13 mW continuous wave (cw) interband cascade laser (ICL) custom-made

by the Microdevices Laboratory at NASA's Jet Propulsion Laboratory (Borgentun et al., 2015) (full instrument schematic shown in Figure 3). The laser emission is centered at 2963 $cm^{-1}$, or 3.37 μm. In 2004, a cw ICL operating at 150 K and centered at 3.57 μm was used as the photon source for the ALIAS-II balloon-borne campaign (Christensen et al., 2007). The first cw ICL operating at ambient temperature was demonstrated in 2008 at 3.75 μm (Vurgaftmen et al., 2015).

The laser output beam is collimated within the laser housing, a standard TO-3 can that serves to protect the laser from ambient

air (most crucially from water vapor, which can condense on the cooled laser). A thermoelectric cooler (TEC) within the housing is used to regulate the temperature to around 275 K. The laser wavelength is tuned by changing the current, which is controlled using a custom designed laser current driver (Sayres et al., 2009). The driver is an FPGA-based programmable waveform generator that achieves highly repeatable, precise and linear current ramps. During flight, the laser completed a scan every 5 msec, which includes 10 % laser off-time used to establish the dark voltage signal from the detector. The in-flight laser

scan covered a range of 3 $cm^{-1}$. This corresponds to a current ramp of 310-355 mA. The wide wavelength range was chosen so $H_2O$ could be monitored during flight via its strong absorption feature at 2961.7 $cm^{-1}$.

Light emitted from the laser TO-3 can passes through a polarizer (ISP Optics, POL-3-5-SI-25) and a quarter-wave plate (ISP Optics, WP-Z-Q-3500). The polarizer and quarter-wave plate (shown collectively as (2) in Fig. 3) serve as an optical isolator that prevents laser light from ultimately reflecting back into the laser housing, which can cause abnormal heating as well as

unwanted feedback. Abnormal heating compromises temperature control, which is necessary to ensure the current ramp consistently yields the same wavelength range emitted by the laser. With the optical isolator, any laser light reflected back from the primary ICOS cavity mirror or other optics is blocked. The effectiveness of the optical isolator was demonstrated separately with the laser on the optical bench before being integrated into the full instrument.

The laser beam then passes through a telescope that reduces the beam diameter to 2 mm, then through a $CaF_2$ 3° wedge (ISP

Optics, CF-WW3-12-2). The wedge acts as a beam splitter. Approximately 7 % of the laser light is reflected onto two gold-plated Pyrex steering mirrors and directed through a diagnostic germanium etalon (LightMachinery, OP-5483-76.2) and onto a mercury cadmium telluride (MCT) detector (InfraRed Associates, MCT-4.5-R-1.0) operated at room temperature. The detector is coupled with a pre-amplifier. The etalon is used to establish the tuning rate of the laser, whose wavelength does not change linearly with current. The free spectral range of the etalon at 2963 $cm^{-1}$ is 0.017474 $cm^{-1}$.

The majority of the laser beam passes through the beam splitter and is reflected by two gold-plated steering mirrors (Thorlabs, PF10-03-P01-10). The steering mirrors direct the beam through a hole drilled in the re-injection mirror (RIM), located 2 cm from its center. The RIM is a 7.6 cm diameter gold-plated mirror with a radius of curvature (ROC) of 100 cm (Thorlabs,





CM750-500-M01). The addition of the RIM is a relatively recent innovation that seeks to mitigate a major disadvantage of cavity enhanced absorption techniques: a large loss of laser power resulting from the high reflectivity of the cavity mirrors

(Leen et al. 2014). In the usual OA-ICOS setup, more than 99.98 % of the laser light is reflected away by the first ICOS mirror before entering the cavity. This loss in photon flux was considered necessary to obtain the large effective path length within the cell, leading to a dramatic increase in the percent absorbance associated with molecular transitions occurring in the cavity (Ouyang et al. 2012). With the RIM present, when 99.98 % of the photon flux is reflected away from the cavity mirror, it hits the RIM but is shifted spatially with respect to the entry hole. The laser beam then reflects off the cavity mirror and RIM 10-

15 times before finally passing back through the hole in the RIM, essentially forming a Herriott cell between the RIM and the first cavity mirror. The result is that more of the light enters into the cavity with each pass—hence why it is referred to as a 're-injection' mirror—and ultimately increases the laser power that impinges on the detector.

The HCl ICOS cavity mirrors (LohnStar Optics, custom) are separated by 47.37 cm. Each mirror is made of zinc selenide (ZnSe) and has a highly reflective coating on the concave side facing inward on the cell, forming the high-finesse cavity. Each

mirror also has an anti-reflective coating on the plano side facing away from the cell. The light is reflected multiple times between the cell mirrors of an ICOS cavity, creating an effective average path length that is a factor of $1/(1 - R)$ longer than the actual pathlength (Ouyang et al. 2012). The mirrors have a light loss of 200 ppm at 3.34 μm (R = 0.9998, or 99.98 % reflective), which is determined by pulsing light into the cell and measuring the e-folding time for decrease in light intensity (the cavity time constant). For this HCl ICOS instrument, the effective path length within the optical cavity is on average 2.368

km. The cavity mirrors both have diameters of 5.1 cm (2 in.). However, they are asymmetric in that they do not have the same ROC. Their ROCs are 200 and 1000 cm, where the mirror with smaller ROC is located on the laser side of the cell. This mirror configuration was chosen to focus the light as it leaves the cavity on the detector side, in order to increase the fraction of light impinging on the detector, as opposed to a standard confocal geometry where light leaving the cell is divergent.

Upon exiting the cavity, the laser beam passes through two positive meniscus lenses (ISP Optics, ZC-PM-50-76 and GE-PM-

12-25-C-2) that focus the light onto the detector. The first focusing lens immediately follows the detector-side ICOS mirror. This lens is 5.1 cm (2 in.) in diameter with a focal length of 7.6 cm. The second focusing lens has a 1.3 cm (0.5 in.) diameter with a focal length of 2.5 cm. This lens is placed about 1 cm away from the detector. During the HUSCE balloon flight, the detector used was a four-stage thermoelectrically cooled MCT detector coupled with a pre-amplifier (Vigo, PVI-4TEMXPXX-F). The TEC cooled the detector to around 193 K, and the detector itself was 1 mm in diameter. The responsivity at this

wavelength is 2.5 A/W. The two-stage pre-amplifier, along with an anti-aliasing filter, applied a gain of $1.02 \times 10^6$ V/A. Throughout the flight, around 130 mV was recorded. A FPGA-based digitizer and processer board using 100 MHz, differential analog-to-digital converters recorded 600,000 raw samples for each laser scan, each scan taking 5 msec. The raw samples were averaged in clusters of 200 to produce 3000 net samples per scan. The scans were then further averaged to 1 Hz and recorded both on the in-flight computer and sent remotely to a ground computer (in case the instrument's landing after flight rendered

the data unrecoverable).





An electronically controlled pinch valve was used to regulate pressure in the cell both in lab and during flight. Air inlet and outlet ports for the cell are both perpendicular to the cell's axis to improve thorough mixing within the cavity. Furthermore, the inlet and outlet are close to the ends of the cell to prevent the formation of dead space near the mirrors. Air is pulled through the system by two scroll pumps each drawing up to 50 standard liters per minute (Scroll Labs, SVF-50P). These pumps were

chosen for their compactness and low weight (two important considerations for airborne campaigns). The net flush rate for the cell is 0.83 times per second.

HCl is a particularly challenging molecule to measure, primarily due to how easily it adsorbs to surfaces. In addition, HCl reacts chemically with surfaces and can be quite corrosive. Thus, the most important aspect of the material that composes the cell and gas flow tubing is that minimal HCl adsorb to its surface. To address this, the cell, the intermediate gas bottle, the

pressure gauge in the gas deck, and nearly all of the plumbing was treated with in an inert silicon coating (SilcoNert2000 coating by SilcoTek), a coating that is considered to have excellent compatibility with HCl specifically (SilcoTek, 2020). The coating is applied via chemical vapor deposition, a process that requires the treated material to be heated to extremely high temperatures. Aluminum consequently could not be used to make the cell and plumbing, despite its advantages of being lighter weight and possessing higher thermal conductivity than stainless steel, the material chosen for this instrument.

Pressure and temperature in the cell are both measured 4 times per second. The cell pressure is measured by an Omega pressure sensor (PX409-005A5V), which was calibrated before flight. The recorded cell temperature is the average reading of three 100 k$\Omega$ thermistors (TDK, B57540G0104F000), located along the cell's interior. During flight, temperature was regulated to 310 K both in the cell and at the inlet. This was done with 5 Minco thermofoil heaters placed uniformly along the cell's exterior. There are two reasons for heating the cell. First, regulating the cell temperature to a constant improves precision by reducing

temperature-related changes in the absorption feature. Second, heating the cell and inlet further reduces the extent to which both HCl and $H_2O$ adhere to these surfaces.

Laboratory assessments of the instrumental performance, which are discussed further in Section 2.3, were based on flowing diluted HCl through the instrument. Stratospheric HCl typically does not exceed 3 ppbv, but due to HCl's corrosive nature and ease with which it adsorbs to surfaces, it is not possible to purchase gas cylinders with HCl mixing ratios this low. The cylinder

used for laboratory assessments of the HCl instrument is $4.6 \pm 0.2$ ppmv HCl in a balance of $N_2$ gas (Airgas), the lowest mixing ratio the company could produce (shown as (10) in Fig. 3).

The mixing ratio of HCl in the cylinder is far too high to directly feed into the instrument for analysis. Therefore, air from this primary HCl cylinder is diluted with air from an ultra-zero air cylinder ($N_2/O_2$ blend, Airgas). This is done by filling an intermediate gas bottle located in the instrument's gas deck. Two separate 1/8" stainless steel lines converge at the gas deck

drawing air from the primary HCl cylinder and the ultra-zero air cylinder, respectively. A manual valve intercepts the line to the ultra-zero air cylinder, and it is kept closed while the intermediate gas bottle is filled to 10-20 psia with gas from the primary HCl cylinder. The intermediate gas bottle is then filled with ultra-zero air up to ~1700 psia. Further dilution can be accomplished after flowing some of the gas through the cell and refilling the gas bottle with ultra-zero air. With this strategy, it is possible to achieve dilution of HCl down to just below 1 ppbv HCl. Air leaves the gas deck via 1/4" inert-coated stainless-



steel tubing that connects to the underside of the inlet tube for the instrument; this connection is perpendicular to the inlet's main entrance for sampling ambient air (inlet not shown in Fig. 3). An electronically controlled valve intercepts this connection and determines whether the instrument is sampling from the intermediate gas bottle or the ambient environment.

## 2.3 Instrument evaluation

Extensive tests were carried out in the laboratory to evaluate and characterize the instrument performance. First, the time
response of the flight instrument to precipitous changes in HCl was determined. The instrument reaches a 90 % settling time in 10 sec, referred to as the 90 % response time. Figure 4 shows the decrease in HCl after 15.5 ppbv HCl flow was shut off and replaced with flow of ultra-zero air. Increase in HCl had a similar response time. The response time is comparable to Webster et al., 1994 and Hagen et al., 2014, who both took similar steps to ensure their instruments were suitable for HCl measurements (Hagen et al. made their cell out of Teflon instead of coating a stainless-steel cell with an inert silicon layer).
Importantly, the response time is significantly faster than instruments that did not take these precautions. Even in cases where instruments incorporated short cavities made of material less resistant to HCl adsorption (specifically nonporous fluoropolymer), 90 % response times have been reported to be greater than 90 s (Roberts et al, 2010).

The instrument was also assessed for linearity in its response to different amounts of HCl. This was done by first filling the intermediate gas bottle with gas from the primary HCl cylinder and ultra-zero air as described. At this point, a stepwise dilution
was performed. At each step, around 40 % of the air was passed through the cell (typically around 800 psia). The intermediate gas bottle's pressure was recorded for 1-2 minutes, then refilled back to ~1300 psia with ultra-zero air. The intermediate gas bottle was then closed off, and the pumps ran on the system for around 20 minutes. This was to ensure that no HCl lingered in the plumbing path and to allow the intermediate gas bottle's pressure and temperature to re-stabilize. These steps were repeated 5-6 times, ultimately yielding sub-ppbv levels of HCl. The cell and intermediate bottle were heated to 315 K and 305 K,
respectively. Expected values were determined by using the pressure readings in the gas bottle to calculate the extent of dilution for each sequence. Pressure readings were only used in the calculation once the gas bottle temperature returned to 305 K after being partially depleted or partially refilled.

The HCl mixing ratio for the very first dilution was determined based upon measurements with the instrument using the published spectral parameters from the HITRAN database, which have a combined uncertainty of less than 5 % (Gordon et al.
2017). The mixing ratio of every subsequent dilution was determined using the initial concentration and the ratio of the gas bottle final pressures prior to and after refilling. The results in Figure 5 are the combined data of two separate executions of the dilution strategy described above. The vertical error bars are 1 σ precision for the 1 s readings, and each point is an approximately 1 minute average. (The intermediate gas bottle is relatively small, and only a portion of the gas inside can be flowed for each data point. This made long averaging periods infeasible). The horizontal error bars are the combined
uncertainty of the mixing ratio in the primary cylinder and of the pressure readings. The regression coefficient for the results is 0.98, and all values are within 10 % of the 1:1 correlation line, similar to the performance of past HCl spectrometers (multi-pass, Scott et al. 1999; CRDS, Hagen et al. 2014).



A distinctive feature of the HCl instrument described here is the coupling of OA-ICOS with the RIM. Past literature has credited the RIM with increasing power impinged on the detector by a factor of 22.5 (Leen et al., 2014). It was confirmed that

a similar increase in power was possible for this HCl instrument. However, this did not result in the lowest fractional noise. The laser trajectory was instead aligned to optimize noise reduction. The power increase associated with this alignment was a factor of 9.

It was found that the presence of the RIM did not just increase laser power impinged on the detector; it also reduced the fractional noise by half (i.e. doubling the signal-to-noise ratio, or SNR). This was true across an array of laser scan rates (Figure

6). That fractional noise is not reduced by the same factor that power is increased is due to the fact that optical noise tends to scale linearly with power, while electronic noise tends to remain constant. Figure 6 shows that not all of the noise is scaled up, so RIM leads to a direct improvement in the instrument's SNR. Figure 6 also shows that as the laser scans more quickly across roughly the same wavelength range, the fractional noise is reduced. This is due to the quenching of optical noise created by standing waves in the cavity, whose constructive and destructive interferences translate into oscillations in the signal. Scanning

too fast can begin to compromise the absorption features as well, though, so a balance must be met to achieve optimal signal-to-noise (Witinski et al., 2010).

Following the HUSCE campaign, the TEC MCT detector was replaced with a Stirling-cooled indium antimonide (InSb) detector (Teledyne Judson Technologies, J10D-J508-R02M-60). The InSb detector has a peak responsivity of 3.68 A W$^{-1}$, and its relative responsivity at 3.34 µm is around 70 %. A two-stage pre-amplifier and anti-aliasing filter collectively adjusts the

gain to 5 x 10$^5$ V A$^{-1}$. The InSb detector is cooled to 83 K and has a diameter of 2 mm, both of which are improvements over the MCT detector used in flight. The cooler temperature reduces thermal electronic noise. The larger detector size allows more light to be captured upon exiting the optical cavity.

One downside is that increased detector size tends to increase electronic noise. The InSb detector is 4 times larger than the in-flight MCT detector. As electronic noise tends to scale up with the square root of the active area, the InSb detector should have

around twice as much electronic noise (Hamamatsu, 2011). The electronic noise of the two detectors can be compared by evaluating the standard deviation in current (A) they produce when the laser is off (i.e. no light impinged on the detector and ignoring their respective gains, V A$^{-1}$). When compared directly, the InSb detector's electronic noise was found to be 1.6 times that of the MCT detector. This factor is likely lower than 2 because the cooler temperature offsets some of the added noise caused by the increased detector size.

With the laser on, the increased electronic noise is more than offset by increased power captured by the detector. To illustrate this, fractional electronic noise was used to compare the two detectors. The fractional electronic noise was determined by calculating the standard deviation of the 1 Hz detector signal when the laser was off and dividing that number by the mean signal when the laser was on. The improvement offered by the current configuration was evaluated by comparing the average fractional electronic noise achieved in the laboratory with the new InSb detector with that obtained with the MCT detector in

flight. The fractional electronic noise with the InSB detector was found to be 4.5 times lower than that of the MCT detector





(0.02 % vs 0.09 %), indicating future campaigns with the instrument should expect improvement in SNR from fractional electronic noise reduction alone.

The 30 s precision for the HCl instrument with the InSb detector is 26 pptv (Figure 7). For the 1 hour sample period, the improvement in the SNR follows the theoretical white noise limit ($1/\sqrt{n}$, where n is the number of samples being averaged)

evident in the Allan variance plot. The gas handling system used to obtain these data was modified from the schematic shown in Fig. 3 to enable longer sample times than allowed by the small cylinder located within the gas deck. For these tests, a manually regulated flow from the primary HCl cylinder was mixed with ultra-zero air passed through a 10 SLM mass flow controller. The ultra-zero air flow was increased to supplement the HCl flow until a pressure of 50.5 hPa was attained. The uncertainty in accuracy is 7 % (5 % from uncertainty in spectral parameters; 2 % from uncertainty in ICOS mirror reflectivity

and HCl-surface interactions). In-flight accuracy is given a conservative upper bound (10 %) due to less stable in-flight conditions. The uncertainties of the flight and new configurations as well as other instruments discussed are summarized in Table 1.

## 3 HUSCE flight overview

On the morning of August 24, 2018, a 7 million ft$^3$ helium-filled balloon carrying the payload was launched from the NASA

Columbia Scientific Balloon facility in Fort Sumner, NM. The HCl instrument was secured within a sealed pressure vessel and mounted to a gondola suspended below the balloon. The pressure vessel, filled to 1 atmosphere pressure before launch, ensured the instrument operated under pressure and temperature conditions representative of the laboratory, and was not exposed to the reduced pressures and temperatures of the stratosphere. The pressure vessel was wrapped with highly reflective material to minimize solar heating. The HCl gas deck and an independent ozone instrument were supplied with heaters and

insulated to provide thermal regulation in flight. Figure 8 shows balloon altitude as a function of time for the HUSCE flight. The full length of the flight, from initial launch to release of the instrumental platform from the balloon, was ~5.5 hours (5:35:16). To preserve battery life, the instrument started recording spectra only once the balloon reached its peak height. (Because of the potential for interference from the balloon wake on ascent, balloon-borne measurements are typically only acquired on descent). The HCl instrument operated for a little over 3 hours (3:11:06). The instrument was powered on for 26

minutes prior to recording spectra to 1) allow time for system controls to stabilize, and 2) record ringdown spectra to determine the reflectivity of the ICOS mirrors, R = 0.9998.

The descent rate of the balloon was adjusted in real time. Two alternate methods were employed: 1) helium was released from the balloon to increase the descent rate, and 2) ballast stored on the gondola was released to decrease the descent rate. These competing controls allowed the balloon to level off just above the tropopause near the end of the flight (Fig. 8). The tropopause

height was determined using the ambient temperature and altimeter readings made on the gondola. The gondola was detached from the balloon and began descent back to the surface at 17.8 km. A parachute was deployed to slow the descent of the gondola, and crash pads located on the underside of the gondola helped absorb the shock upon landing.



### 3.1 Flight-specific instrument details

During flight, a pinch valve was used to regulate cell pressure at 53 hPa. At ambient pressures below 60 hPa, the pinch valve
remained completely open to maximize flow through the cell. The cell pressure gradually rose from a minimum of 16 hPa at
maximum altitude (29.5 km) to the point where the pinch valve started regulating to maintain 53 hPa (19.5 km).

Ambient pressure was measured during flight by an Omega pressure sensor (PX409-015A5V) attached to a port on the side of
the pressure vessel. The sensor was calibrated before flight and also verified against ambient pressure readings collected by
the Columbia Scientific Balloon Facility at Fort Sumner. The zero offset of the on-board pressure sensor was adjusted as a
result of this comparison. The NASA facility also measured ambient temperature as well as altitude via GPS. These
measurements were used solely for describing the flight profile but were not used in the formal analysis of the dataset obtained
from the HUSCE flight.

Water vapor was also measured by the HCl instrument during flight. The laser scan range includes a strong water vapor
absorption feature at 2961.7126 cm$^{-1}$ (Figure 2). Background water vapor present in the pressure vessel distorted the water
vapor absorption feature, however, and limited the accuracy and precision of this measurement. This interference arises as a
result of the absorption path outside the cell between the RIM and laser-facing ICOS mirror. Because the extra-cavity volume
within the pressure vessel is held at ~1000 hPa, the water vapor interference is pressure-broadened relative to the signal present
within the sample cell (≤53 hPa) and primarily impacts the baseline of the ambient signal. Thus, despite this background
interference the H$_2$O measurements can be used for diagnostic purposes, in particular, to identify contamination of the ambient
environment from the balloon and gondola. Large and rapid changes in sampled water vapor are evident on top of a nominally
constant concentration of background water vapor in the pressure vessel.

### 3.2 Flight HCl data processing

While the average laser power at the detector remained stable throughout the flight, intermittent power oscillations with a
frequency of 7.9 kHz and variable amplitude, were observed in nearly half of the recorded spectra. Laboratory assessments
after the campaign suggest the origin of the oscillation was electronic and likely due to a grounding issue on the gondola itself.
In the following analysis, the 1 s spectra were fit then averaged in 30 s bins. The baseline was fit with the absorption feature
using a least-squares fitting algorithm described in detail in Sayres et al., 2009 and Allen, 2020. Briefly, the laser power curve
and electrical oscillations (if present), along with minor optical etalons that formed in the cavity are modeled using a fourth-
order polynomial and three sine and cosine waves. The use of sine and cosine waves for each frequency allows the algorithm
to account for the phase of the oscillation. The absorption feature is fit using a Voigt line-shape function. The spectrum in Fig.
9 (bottom panel) is a 30 s averaged spectrum that corresponds to an ambient mixing ratio of 1.15 ppbv HCl, measured at
atmospheric pressure of 26.3 hPa. The fit residual is shown in the top panel of Fig. 9, and the spectrum itself is shown in the
bottom panel. The ICOS spectra were fit using HITRAN spectroscopic parameters for the HCl transition described in Section





2.1 and converted to mixing ratio using measurements of cell temperature and pressure. The fitting algorithm itself was written
in C++, with supporting scripts written in MATLAB R2019b.

The 30 s average spectra had variable precision throughout the flight, in large part due to the sporadic electrical interference which was less present later in the flight. The noise-equivalent absorption, used here for 30 s precision, varied from 40-70 pptv HCl. This range was determined by calculating the relative standard deviations of cell pressure, cell temperature, and the fit residual for each of the 30 second spectra taken during the flight (less than 10 % of the noise-equivalent absorption is due to
uncertainty in the cell pressure and cell temperature). All observations were greater than the detection limit (dl), defined as the noise-equivalent absorption for a 30 second average (70 pptv); more than 99 % of observations were greater than *3 x dl*. While averaging for 30 seconds greatly improved the precision of the reported values, the averaging length should not be shorter than this because of the cell's flush rate and the previously discussed 90 % response time of the instrument.

## 4 HUSCE HCl profile and validation

The HCl profile obtained during the HUSCE campaign (Figure 10, grey circles) is consistent with other reported measurements of stratospheric HCl over the U.S. (Froidevaux et al., 2008). Ranges from MLS observations of HCl made on the same day and nearby geographic region are shown as well, demonstrating the agreement between HUSCE campaign observations and satellite measurements. The comparison with MLS is discussed further in Section 4.1. The pump speed was changed during small portions of the flight to evaluate in-flight HCl adsorption on the inlet and cell. We observed a low bias up to 20 % for
HCl mixing ratios during the lowest pump speeds (1/4 the normal operating pump speed), though atmospheric variability may account for some of this change. Periods where pump speed was lowered like this are excluded from the profile shown in Figure 10.

While the HCl profile is more variable in the mid-stratosphere than in the lower stratosphere, instrument diagnostics do not suggest any operational cause for the increased variability. There is evidence that balloon interference may have impacted
portions of the mid-stratospheric descent, based on anomalous readings from the diagnostic water vapor measurement and the ambient temperature measurement (Kräuchi et al., 2016). However, neither the temperature profile nor the points of elevated water vapor correlate with the observed HCl mixing ratios.

### 4.1 HUSCE comparison with MLS

Observations from the HUSCE campaign were compared with measurements from MLS, which is used as the primary
instrument for comparison because it provides continuous and reliable global measurements of stratospheric HCl for the full altitude range of the HUSCE flight.

MLS measures HCl by observing the molecule's rotational transition band at 640 GHz, with an estimated single-profile precision of 0.2 ppbv in the pressure range encountered during the HUSCE flight (Livesey et al., 2020). From stratospheric pressures 100 to 0.15 hPa, MLS agrees well with ACE-FTS (within 5 % on average) and with previously made in situ balloon-
borne observations launched from Fort Sumner, New Mexico (Froidevaux et al., 2008). For comparison with HUSCE, 11 MLS





observations were selected from August 24, 2018 (Figure 11). The range of these observations, including error bars, is shown as green shading in Fig. 10, with their average profile shown in dark red. In this case, and in all cases where MLS data are used for illustration or comparison in this paper, the recommended quality control was applied (Livesey et al., 2020). For more details on MLS, see Waters et al., 2006.

The MLS data between 60 and 20 hPa illustrate that the HCl profile does not have a monotonic increase with altitude. The dashed lines that connect each MLS profile are meant to simply help guide the eye. They should not be understood as an accurate or even reasonable interpolation. Considering the non-monotonic variability in all of the profiles, it is much more likely that HCl mixing ratios at intermediate pressures would have more variation than the linear interpolations suggest. Finally, while the MLS observations closest to HUSCE were chosen for this comparison, the spatial and temporal overlap is

not perfect, and some disagreement is expected. Nonetheless, the HUSCE observations of HCl agree well with MLS, with all of the 5 hPa averaged observations being within range of MLS observations that day and more than 95 % of the 30 s averaged in situ observations being within the MLS error bar range (Figure 10, left panel).

Reported MLS values are the result of averaging mixing ratios at various pressure levels described by the MLS averaging kernel. For HCl, however, more than 98 % of the weighting at the 3 pressure levels shown in the right panel of Fig 10 are at

those exact pressure levels (Livesey et al., 2020). For calculating the percent difference then, all 1 s HUSCE observations within ± 1 hPa of the reported MLS pressure level were binned and averaged. That average was then compared to the mean of MLS observations taken at 8:25:36 and 8:26:40 UTC, the 2 closest observations to the HUSCE trajectory (see Fig. 11 for those individual profiles and location). For the 3 pressure levels at which this comparison between HUSCE observations and MLS was feasible, the average absolute percent difference was 8 %, and all differences were within the relative uncertainty of the

MLS observations (Fig. 10, right panel).

A separate instrument was used to measure ozone during the flight. The instrument, a multi-pass white cell with a UV LED light source, detected ozone by measuring the absorption of UV light at 255 nm. Sample gas was alternately directed into the detection cell to measure the absorption signal or through a $MnO_2$ scrubber in order to measure the background for the determination of ambient ozone concentrations. The measurements generally agree with MLS observations as shown in Figure

12, with similar expected discrepancy as for HCl (due to imperfect agreement on location and time). An ozone sonde profile is also included in Figure 12 and shows better agreement with the HUSCE $O_3$ profile. This profile was observed by a NOAA sonde in Boulder, CO on August 23, 2018 at 17:24:16 UTC (NOAA 2018). Note that ozone continues monotonically increasing between 50-20 hPa for all of these data—unlike HCl.

The difference in behavior between the ozone and HCl profiles observed during HUSCE may be explained by examining

global stratospheric profiles of these two gases. To illustrate this, zonally and meridionally averaged profiles for the month of August 2018 were determined for MLS ozone and HCl from 100 to 5 hPa. The resulting contour plots of MLS HCl mixing ratio as a function of latitude and longitude are shown in Figures 14 and 15, respectively. The HCl mixing ratios are based on the monthly means for August 2018 for each pressure/latitude coordinate (or pressure/longitude in Fig. 14). The HCl mixing ratios are provided by MLS as averages in the following pressure bins: 100, 68, 46, 32, 22, 15, 10, 7, and 5 hPa. The trajectory



of HUSCE balloon descent is represented by a white dotted line in the figures. Temperatures and pressures used to determine the zonally (meridionally) averaged potential temperature surfaces for August 2018 were obtained from the European Center for Medium-Range Weather Forecasts ERA-Interim reanalysis (ECMWF, 2011). The black lines shown in Figures 14 and 15 represent the derived isentropic contours. The calculated potential temperatures ranged between 365 and 630 K at 100 and 20 hPa, respectively. Though not shown, isentropic contours derived from MLS temperature profiles yielded similar results. The

above analysis was also performed for methyl chloride ($CH_3Cl$) (Fig. 15) and ozone (Fig. 16). $CH_3Cl$ mixing ratios are binned by the same pressures as HCl, and ozone mixing ratios have twice as many pressure bins.

The general trend for HCl in the stratosphere is consistent with the degradation via photolysis and oxidation via OH of the organic chlorine compounds (e.g. CFCs and $CH_3Cl$) and their subsequent conversion to inorganic forms (e.g. HCl). This is evident in Figure 14, where the monotonic increase in HCl with altitude is consistent across longitudes. However, Figure 13

illustrates that the trend is not as simple across latitudes, reflecting the influence of both chemistry and dynamics (e.g., stratospheric circulation and mixing) on the distribution of long-lived trace species such as HCl. For much of the stratosphere, HCl has lower levels as one nears the equator. Between 10-50 hPa a strong gradient in HCl with latitude is evident. Figure 13 also shows that this meridional gradient is even more pronounced than the vertical gradient for this pressure range and latitude. The figure further shows that the isentropic surfaces are relatively flat in this pressure range and cut across the gradient in HCl.

Figure 15 is the same as 13 except the molecule plotted is $CH_3Cl$. $CH_3Cl$ is the dominant natural source of chlorine in the stratosphere, providing around 16 % of stratospheric chlorine in 2000 (Waters et al., 2006). By comparison, CFC-11 contributes around a quarter of the stratospheric chlorine budget, despite observed declines after being largely phased out (Montzka et al., 2018). The $CH_3Cl$ profile gives some indication of the rate at which chlorine radicals are formed from tropospheric air that has been uplifted into the stratosphere. The stratospheric profile of $CH_3Cl$ is roughly the inverse of that

of HCl, demonstrating the chemical conversion from organic to inorganic forms of chlorine as air ages in the stratosphere.

Figure 16 shows the global profile across latitudes for ozone. Ozone also has some latitude dependence. However, its change with latitude is significantly less pronounced than that of HCl and $CH_3Cl$. There is virtually no horizontal gradient near the latitude at which HUSCE platform descended in the stratosphere. Additionally, the subtle latitude dependence that does exist near the HUSCE observations is roughly confined within the isentropes, suggesting that the ozone instrument on the HUSCE

campaign should have observed a monotonic decrease of ozone as it descended—even as it crossed through different isentropes. This is consistent with our observations. Importantly, this expectation is in stark contrast with the HCl profile, both in our measurements and in the MLS August 2018 average.

### 4.2 HCl correlation with ozone

In the lower stratosphere, observations made with a chemical ionization mass spectrometer (CIMS) that was flown aboard

NASA's WB-57 aircraft demonstrated that HCl has tight correlation with ozone (Marcy et al., 2004). CIMS was deployed to measure HCl in the lower stratosphere as part of the Cirrus Regional Study of Tropical Anvils and Cirrus Layers-Florida Area Cirrus Experiment (CRYSTAL-FACE) mission out of Key West, FL, in the summer of 2002, and again as a part of the Aura





Validation Experiment (AVE) out of Houston, TX, in 2005. (data available at NASA, 2005; results published in Froidevaux et al. 2008).

Marcy et al., 2004 established that a tight correlation exists between HCl and $O_3$ in the lower stratosphere. Regression analysis by Marcy et al., 2004, from the CRYSTAL-FACE mission in late July yields slopes of $0.00044 \pm 0.00004$ and $0.00051 \pm 0.00004$ for two independent flights taken days apart, covering latitudes 24-27° N and 24-39° N, respectively. Observations between 30-40° N and at ambient pressures lower than 80 hPa from the AVE mission, taken using the same instrument, have a slope and standard error of $0.00074 \pm 0.00003$. HUSCE HCl and $O_3$ measurements in the lower stratosphere (65-80 hPa)

yield a slope of $0.00055 \pm 0.00007$ (Figure 17). The differences in slopes obtained by these different campaigns illustrates the differences in stratospheric chemistry on $O_3$ and HCl, and therefore their correlation, with latitude and season as discussed in section 4.1.

**5 Discussion and conclusion**

In this paper, we report on a new RIM-ICOS instrument that measures atmospheric HCl by probing the molecule's fundamental

ro-vibrational transition at 3.37 μm. The instrument responds linearly to changes in HCl, as shown by the described dilution technique that assessed linearity down to sub-ppb levels. The instrument has a demonstrated response time of 10 s, significantly faster than commercial ICOS instruments used to measure HCl and comparable to another cavity enhanced HCl spectrometer (Hagen et al., 2014).

We also evaluated the instrument performance during the HUSCE campaign, a balloon-borne descent made over the central

U.S. during August 2018. HUSCE's HCl observations agree well with MLS observations made that same day. Vertically averaged values are 8 % different from MLS on average, all within MLS reported uncertainty. Finally, HUSCE's HCl measurements and $O_3$ observations produce a tight regression coefficient within range of previously reported values. Overall, the results of the HUSCE engineering test flight demonstrate that the HCl instrument performs well under flight conditions.

Since the HUSCE campaign, an updated detector has been installed in the HCl instrument, and the wavelength range has been

narrowed to focus on the HCl absorption feature. Future campaigns that include the instrument will likely house a separate instrument that measures water vapor. With these improvements, we find the lab-based 30 second precision is 26 pptv.

This is the first time that stratospheric measurements of HCl have been made with ICOS, where even the in-flight, 30 second precision of 70 pptv compares favorably with most other measurement techniques for HCl. This is also the first time any cavity enhanced HCl instrument has been tested in-flight (in contrast to the ground-based field test performed by Hagen et al. 2014).

Finally, all known previous research discussions of RIM-ICOS were either strictly theoretical or coupled with lab-based demonstrations. The HUSCE campaign demonstrates that RIM-ICOS is feasible for making in situ measurements in balloon-borne field deployments. Cavity enhanced techniques, such as ICOS and CRDS, offer the ability to obtain a higher level of precision in a small cell than instruments that rely on direct absorption from multi-pass cells of the same length. Furthermore, the ability of the RIM to enhance signal-to-noise and its stability during flight support integration of this additional mirror for

further gains in precision for in situ instrumentation.



**Data Availability.** The data generated and analyzed for the current study are available at: Wilkerson, J. Harvard University Stratospheric Chemistry Experiment data set, *Harvard Dataverse, V1*, https://doi.org/10.7910/DVN/PMLPRR, 2021.

**Author Contributions**

D.S.S., J.B.S., J.W., and J.G.A. designed the study. M.R., M.G., N.A., J.W., and D.S.S. developed the HCl instrument. J.B.S. developed the ozone instrument that flew alongside the HCl instrument and processed its data. All authors contributed to data collection and field work during the HUSCE campaign. J.W., D.S.S and N.A. contributed to data processing, lab testing and uncertainty analysis for the HCl instrument. J.W. was mainly responsible for interpreting the results and writing the manuscript. All authors participated in editing the manuscript.

**Competing Interests**

The authors declare that they have no conflict of interest.

**Acknowledgements**

We thank Columbia Scientific Balloon Facility for providing the resources to evaluate the instrument performance in the stratosphere. This work was funded by NASA grant NNX16AI72A.

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

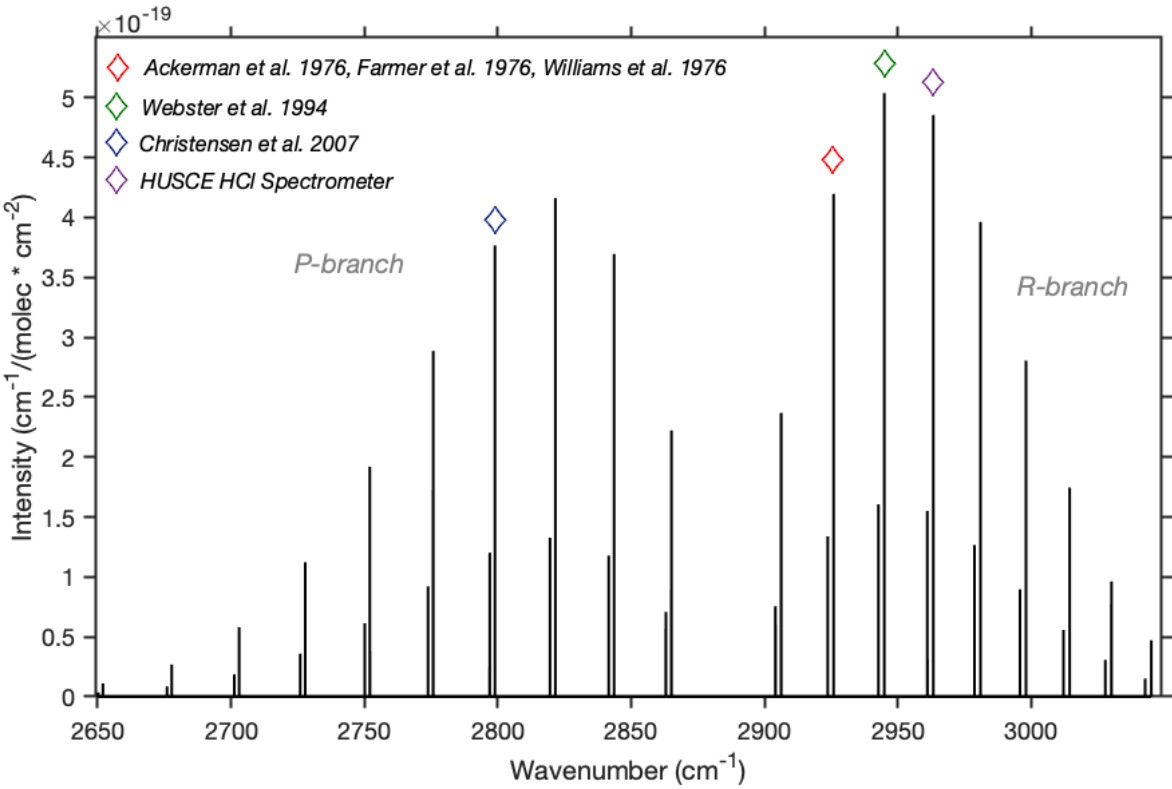

**Figure 1**: Line intensities of fundamental transition of HCl. The relatively smaller line that accompanies each larger line represents transition of $H^{37}Cl$. The purple diamond indicates the transition that the current instrument uses to measure HCl. The red, green, and blue diamonds indicate transitions utilized by past measurements of stratospheric HCl.





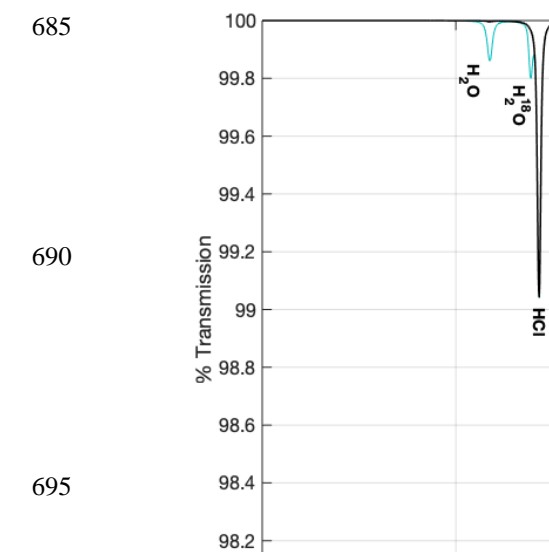




**Figure 2**: Theoretical spectrum in vicinity of HCl feature. Parameters are P = 53 hPa, T = 310 K, HCl = 1.0 ppbv, and $H_2O$ = 4.5 ppmv (black) and 200 ppmv (blue). P and T are in the range of what was observed in-flight, and mixing ratios in black are representative of the stratosphere.




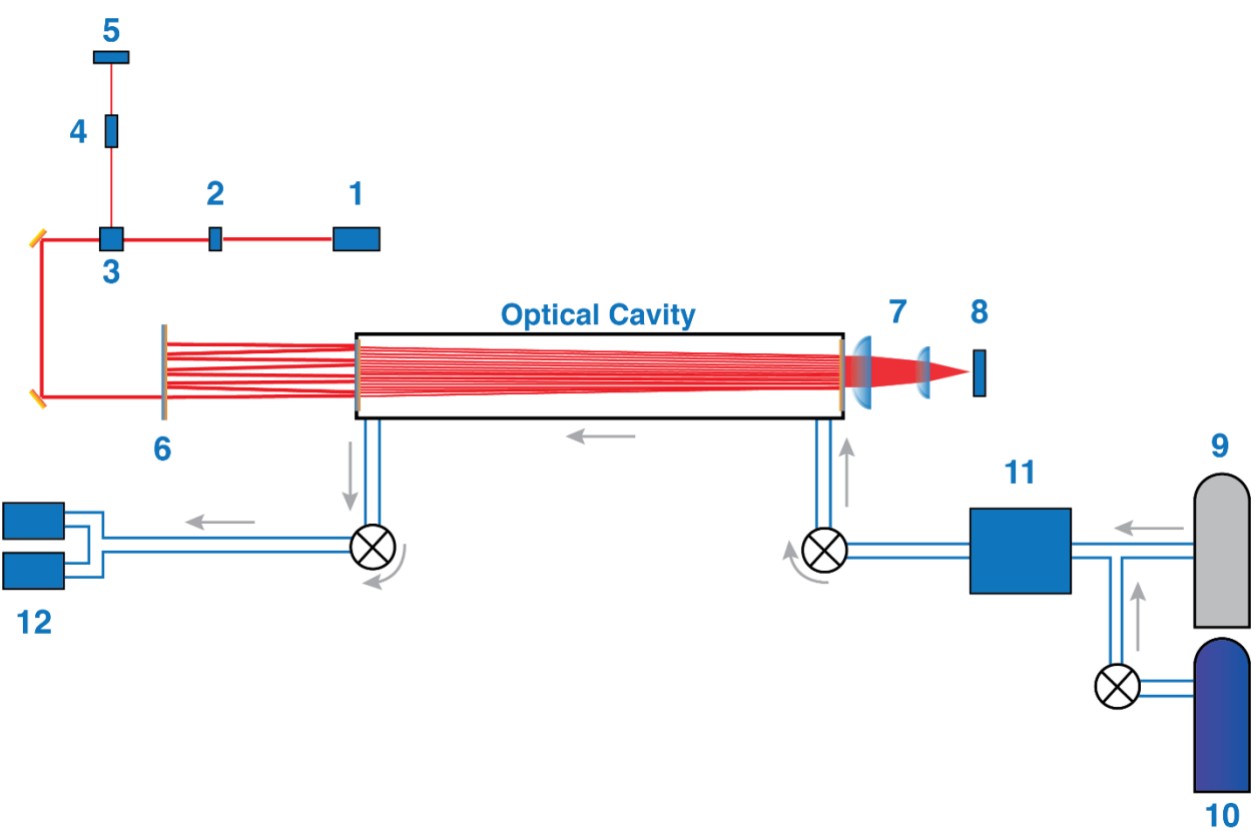

**Figure 3**: Diagram of the HCl instrument layout. The schematic is intended to clarify the path of laser light and gas flow path in a lab setting (optical setup is identical in flight) 1) ICL and laser TEC, 2) optical isolator, 3) beam splitter, 4) etalon, 5) etalon MCT detector, 6) RIM, 7) focusing lenses, 8) ICOS detector, 9) tank of ultra-zero air, 10) tank of 5 ppmv HCl, 11) gas deck, which includes the intermediate gas bottle, and 12) vacuum pumps. Grey arrows indicate gas flow path. The main text describes each component in more detail.







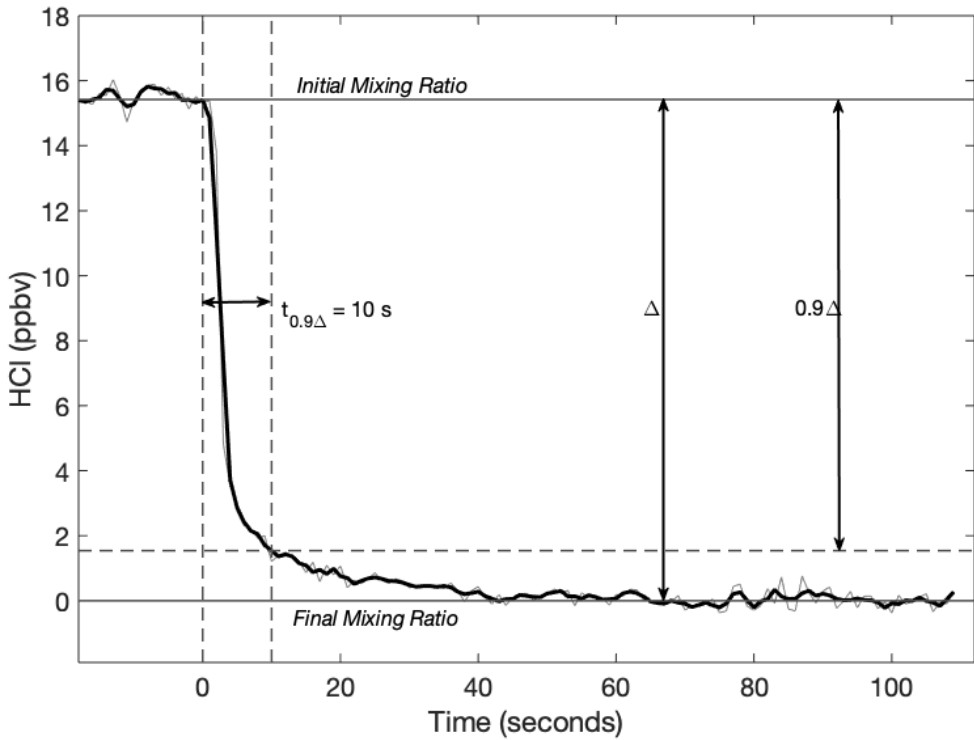

**Figure 4**: Response time of HCl instrument when 15.5 ppbv HCl was replaced with ultra-zero air flow. The observed mixing ratio is 10 % of the input value after 10 s. The light grey line is 1 Hz sampling. The black line is a 3 s smoothing average. The cell was heated to 315 K, and cell pressure was 50.7 hPa.







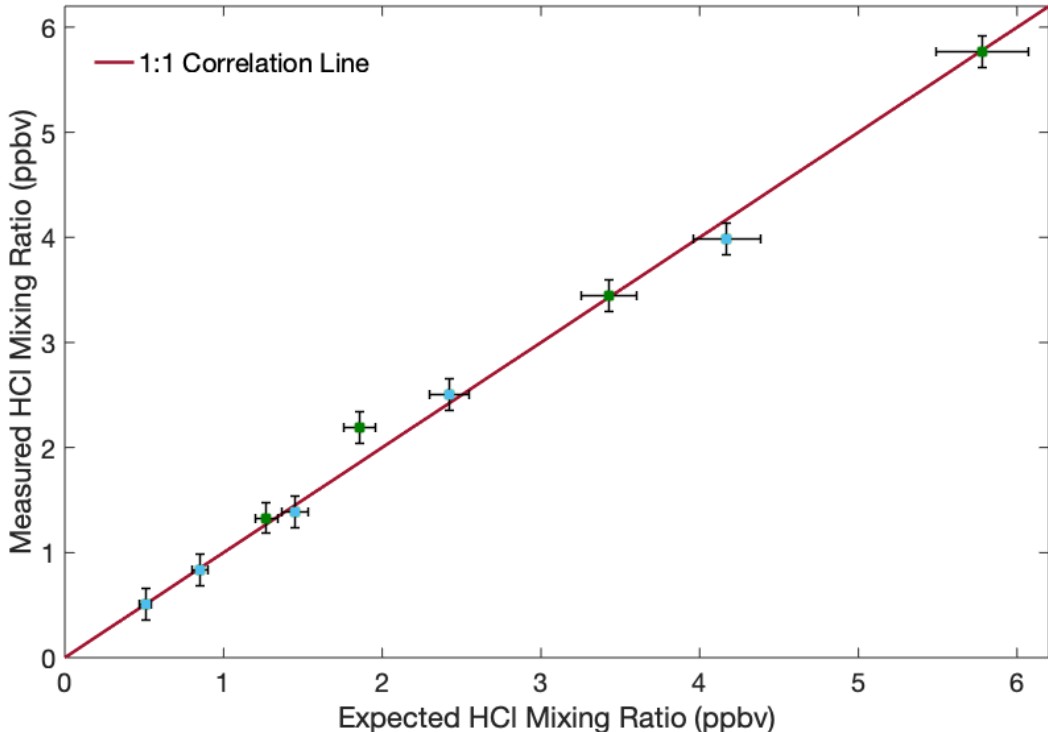

**Figure 5**: Observed HCl mixing ratio after sequential dilution of original HCl cylinder to illustrate that the instrument responds linearly to different input mixing ratios. Each point represents an approximately 1 minute average of 1 Hz spectra. The cell was heated to 315 K, and the cell pressure was around 24 hPa. Vertical bars represent 1 σ precision for the instrument readout. Horizontal bars represent the combined uncertainty of the input HCl cylinder and the pressure readings used to quantify dilution. The blue and green points indicate the two separate iterations of the experiment.



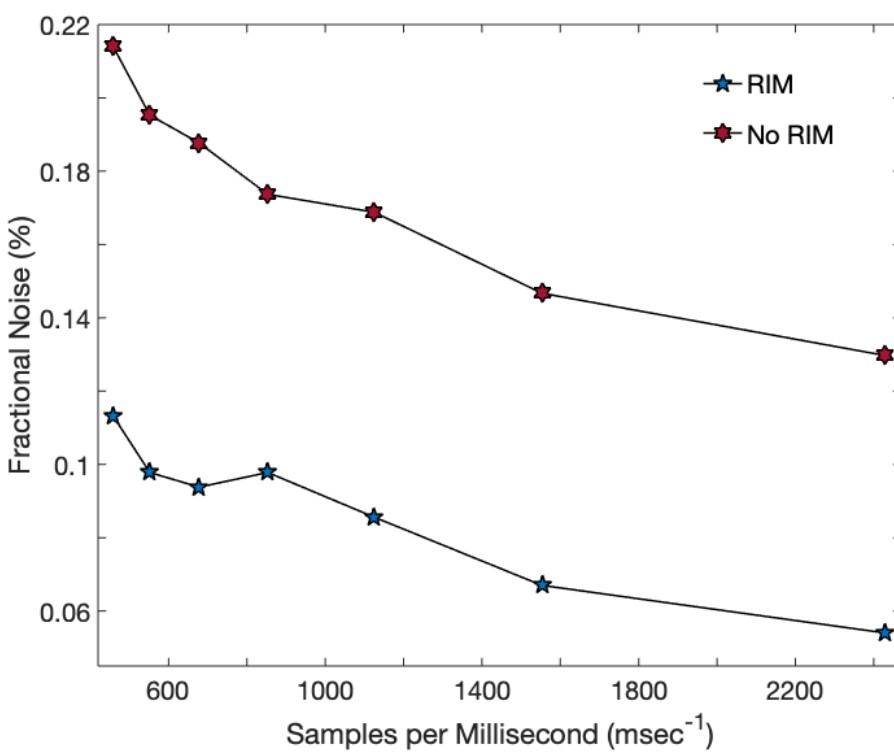

**Figure 6**: Fractional noise as a function of scan rate of the laser. Number of samples per scan was held constant, so larger number of samples observed by msec indicates higher scan rate. Two important trends are illustrated: total fractional noise is lower with faster scan rates, and fractional noise is lower when RIM is incorporated into optical layout.



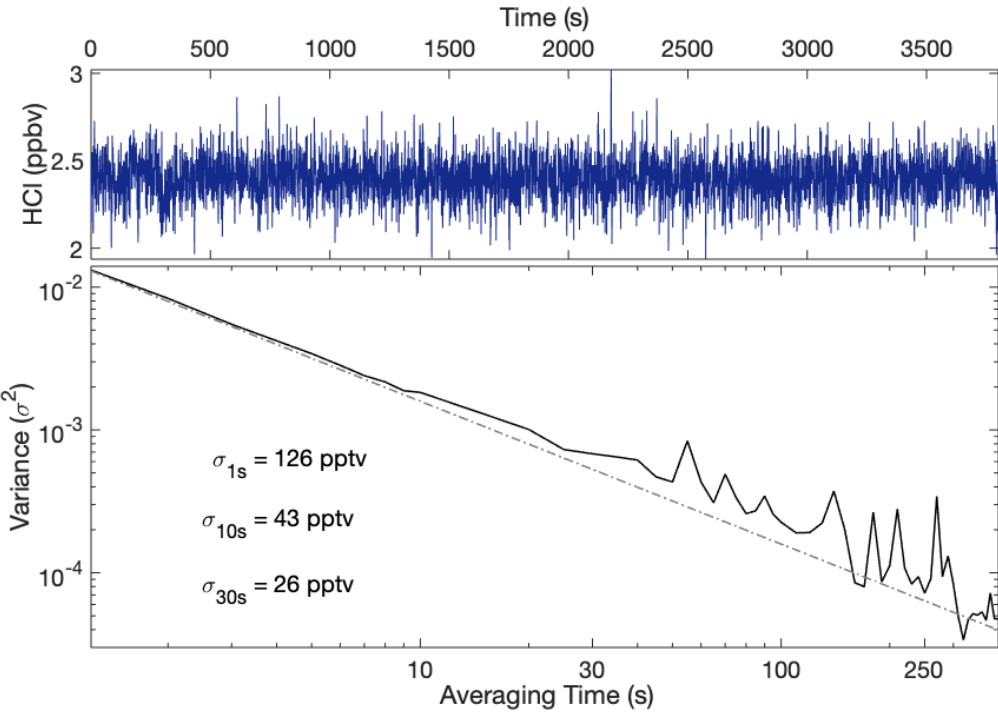

**Figure 7**: Allan variance plot of HCl instrument after ~1 hour of constant flow of diluted HCl. The top panel shows the mixing ratio of HCl over time. The bottom panel shows the variance as a function of averaging time (solid black line) along the theoretical white noise limit (dashed black line), where averaging times up to 7 min. are shown.






**Table 1**. Comparison of uncertainties among HCl instruments.

|  | This work w/ MCT | This work w/ InSb | CRDS[1] | CIMS[2] | ALIAS-II[3] | MLS[4] |
|---|---|---|---|---|---|---|
| Platform | Balloon | Laboratory | Ground | WB-57 | Balloon | Satellite |
| Technique | ICOS | ICOS | CRDS | CIMS | Multi-pass IR spectroscopy | GHz spectroscopy |
| Precision | 70 pptv (30 s) | 26 pptv (30 s) | 20 pptv (60s) | 15 pptv (1s) | 90 pptv (50s) | 200 pptv |
| Accuracy | 10% | 7% | < 10% | 25% | 20% | 200 pptv |
| 90% Response Time | 10 s | 10 s | 10 s | Not reported[‡] | Open path | N.A. |

[1]Hagen et al. 2014; [2]Marcy et al. 2005; [3]Christensen et al. 2007; [4]Livesey et al. 2020

[‡]The response time is not explicitly stated, but the reported accuracy does account for uncertainties associated with the inlet path flow.







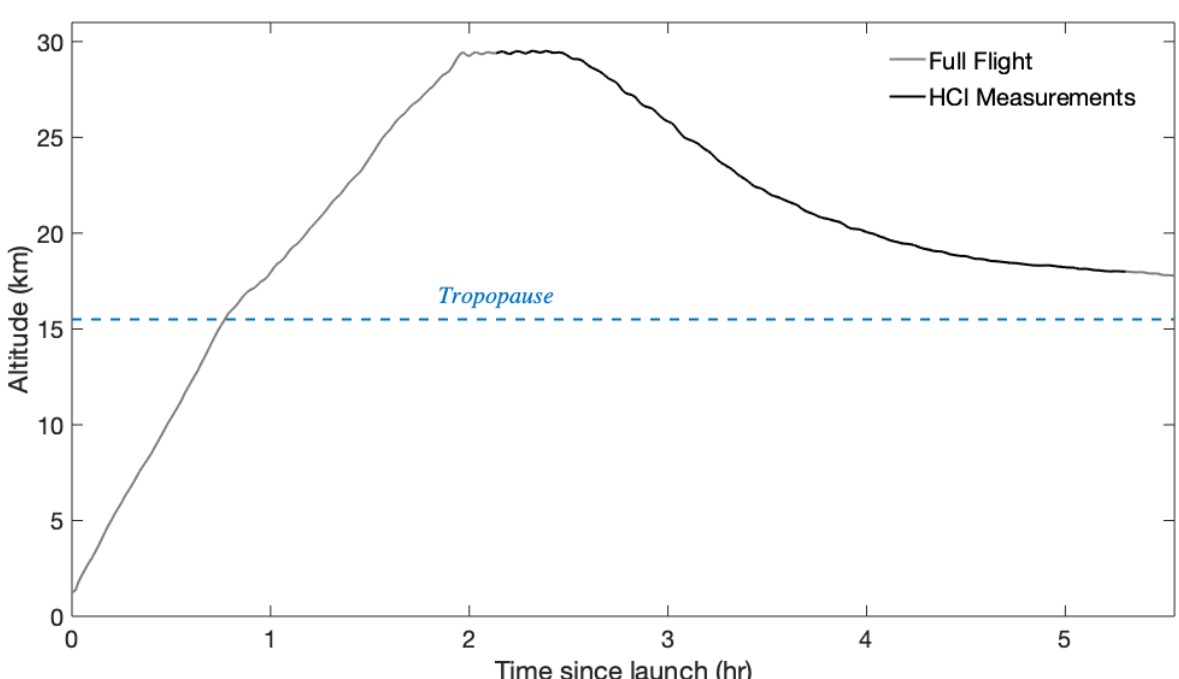

**Figure 8**: The full altitude profile of the HUSCE flight is shown in grey. The black segment of the profile is when the HCl instrument was on and recording measurements. Tropopause determined by observed reversal in lapse rate from ambient temperature sensor on the gondola.

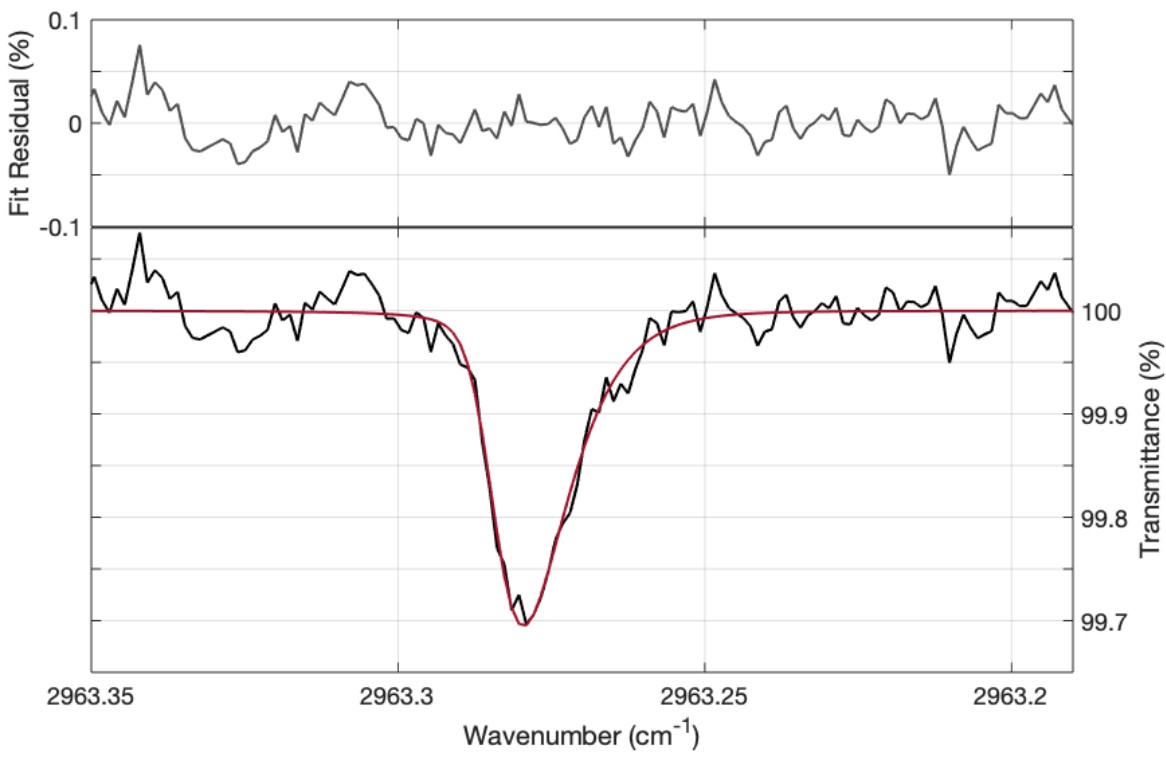

**Figure 9:** Example of a 30 s spectrum from flight, focused on the region that was used to determine HCl mixing ratios. The spectrum corresponds to 1.15 ppbv HCl at a cell pressure of 13.6 hPa. (bottom panel) The black line is the detrended spectrum, and the red line is the fit. (top panel) The fit residual, which equals detrended spectrum – fit, is shown in dark grey. The absolute residual averages to 0.016 % in this example.


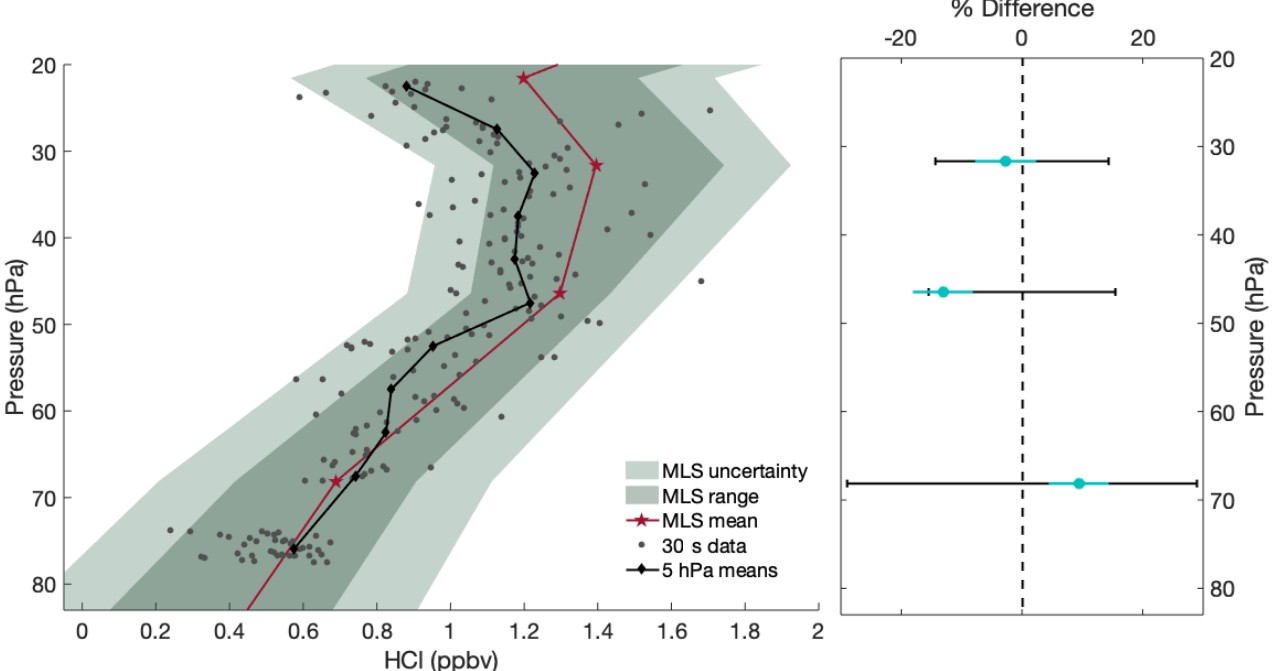

**Figure 10:** (left) Profile of HCl observed during HUSCE in comparison to nearby MLS observations on Aug 24, 2018. Grey dots represent 30 s HCl data from the balloon, and black diamonds represent the vertical average of these data in 5 hPa bins. Red stars represent the average of 11 nearby MLS observations; the dark green shading shows the range of MLS observations, and the light green shading shows the MLS uncertainty. (right) Percent difference between HUSCE observations and the mean of the closest two MLS observations that day, relative to MLS, are shown as cyan dots. Black error bars represent the relative uncertainty for MLS observations, and cyan error bars represent the uncertainty in accuracy for the HITRAN parameters.



**Figure 11:** MLS observations of HCl made near HUSCE on August 24, 2018. Error bars indicate MLS reported uncertainty. Dashed lines are for readability; they do not indicate known trends between the observations shown as colored triangles. Inset: locations of each MLS observation (colors are coordinated with main panel).

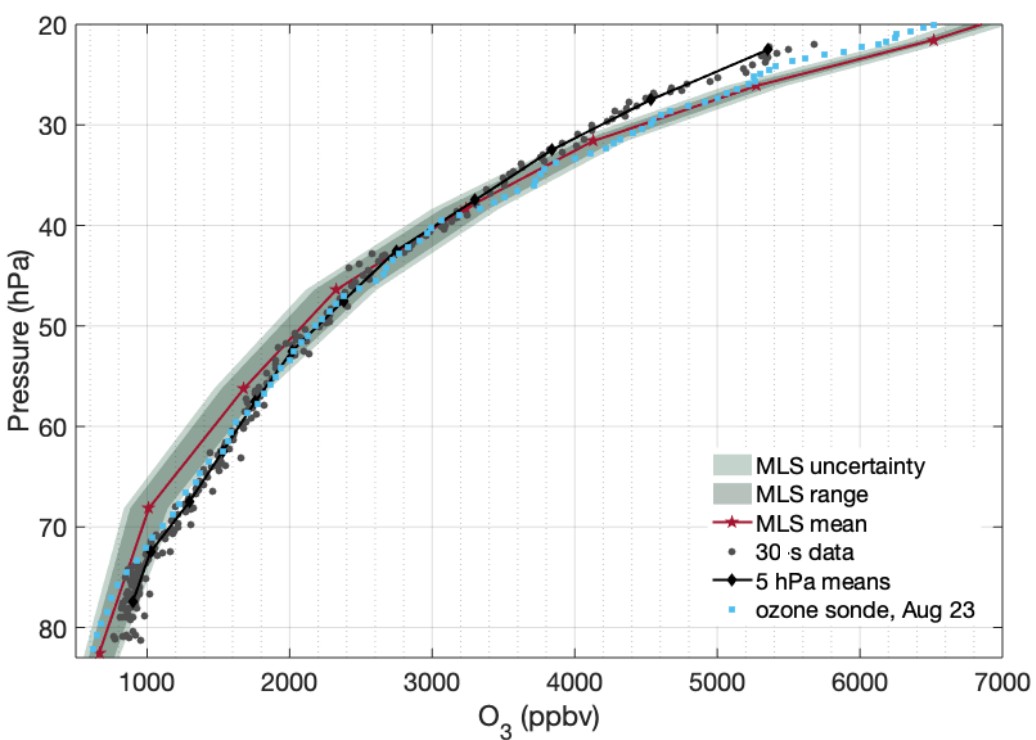

**Figure 12:** Ozone profile from HUSCE (grey circles) overlaid onto 11 MLS observations in the same way as in Figure 10. The 11 MLS observations are at the same locations as those shown in Figure 11. The light blue circles are observations from a NOAA ozone sonde in Boulder, CO on the evening of August 23, 2018.


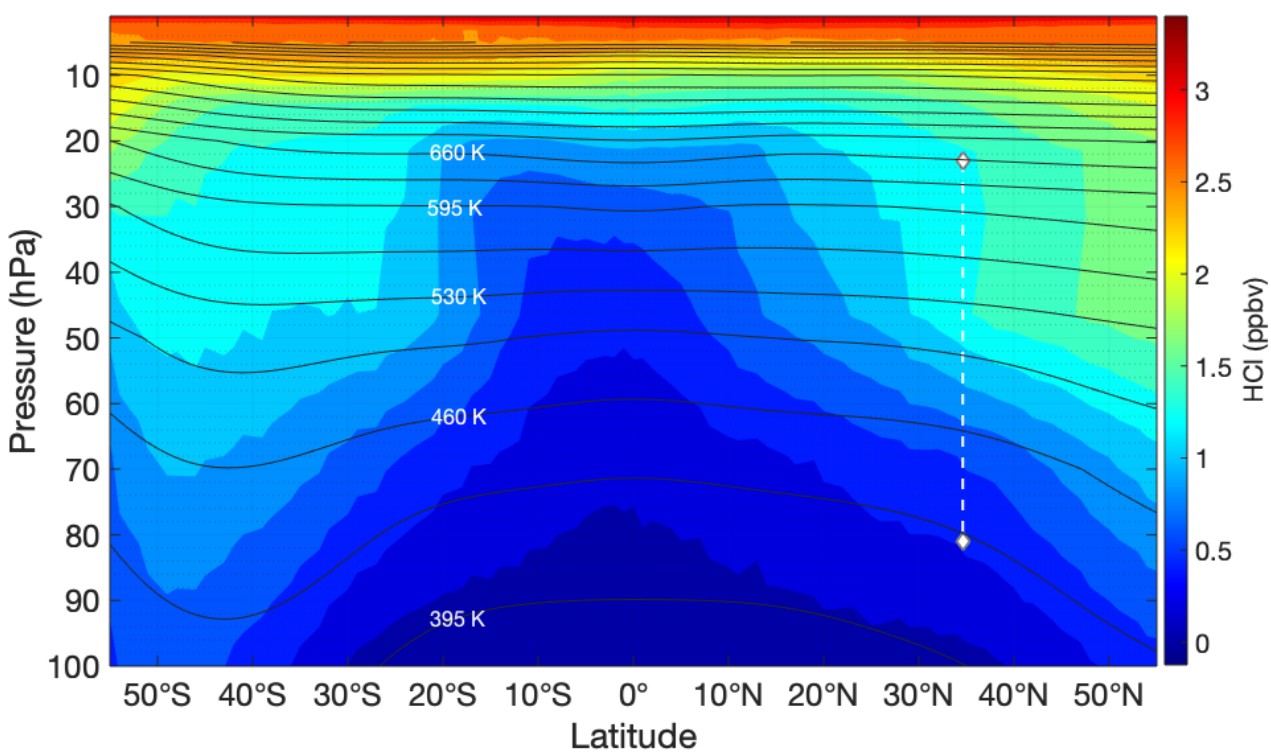

**Figure 13:** Contour plot of zonally averaged MLS HCl profiles as a function of latitude for the month of August 2018. The white dashed line indicates the HUSCE descent. The black lines represent zonally averaged isentropes for August 2018.

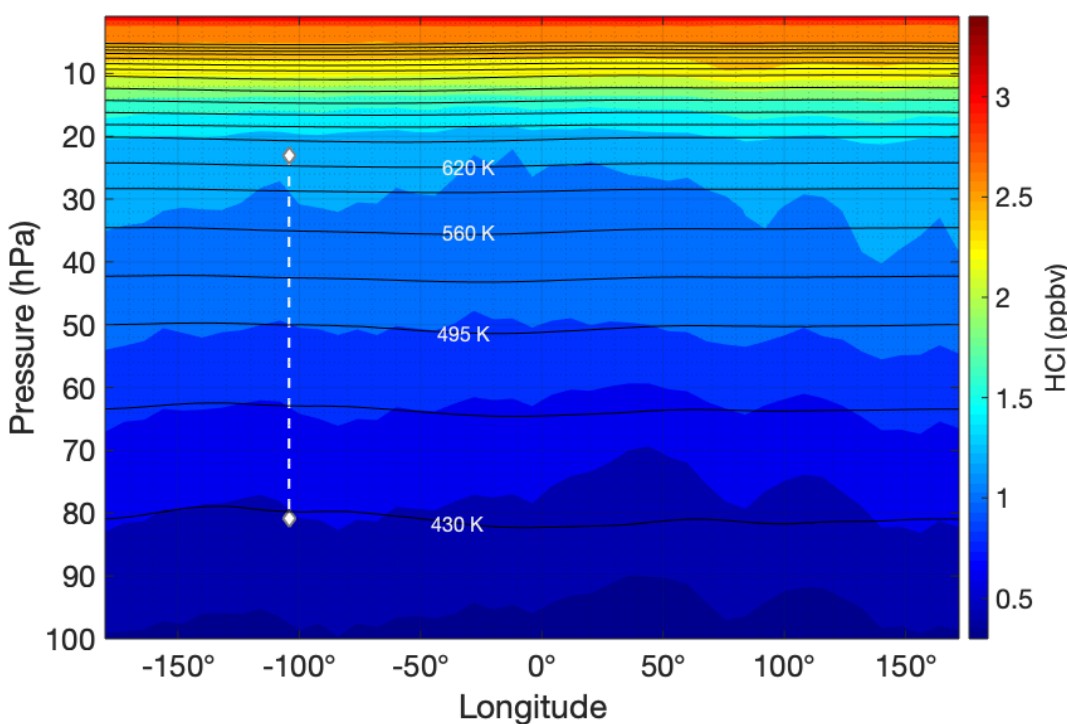

**Figure 14:** Same as for Fig. 13 except the meridionally averaged HCl profile is shown as a function of longitude instead.


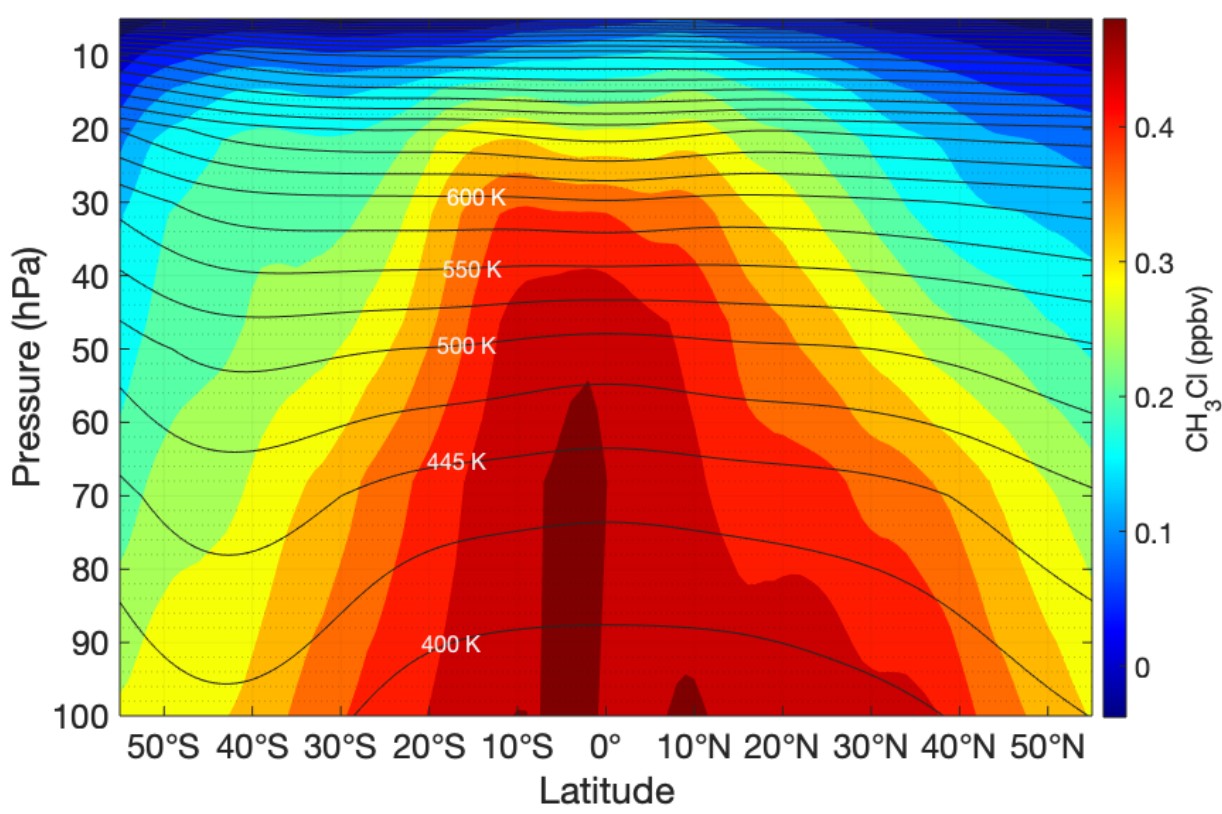

**Figure 15:** Same conditions as for Fig. 13 except the profile illustrated is CH$_3$Cl. Note that CH$_3$Cl has a strong latitude dependence but inverse to the trend seen in HCl.

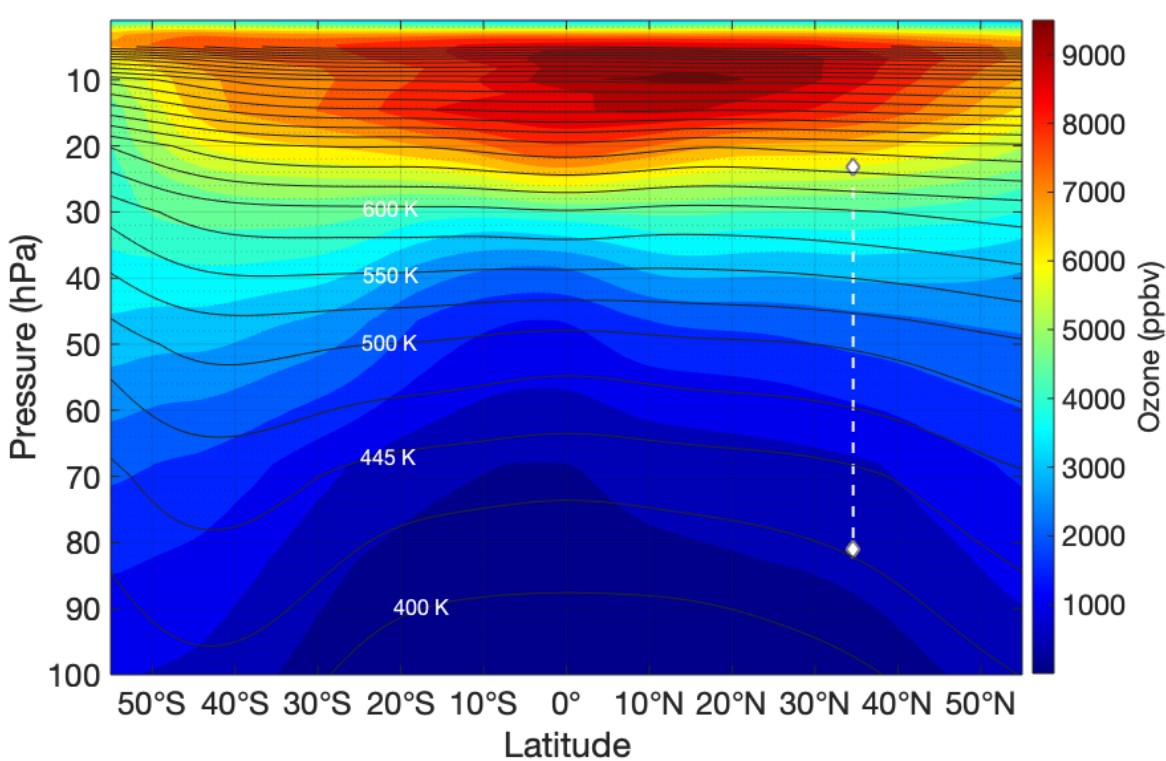

**Figure 16:** Same conditions as for Fig. 13 except the profile illustrated is ozone. Note that ozone has very little latitude dependence at the altitudes and latitude that HUSCE observations were made, especially compared to HCl.





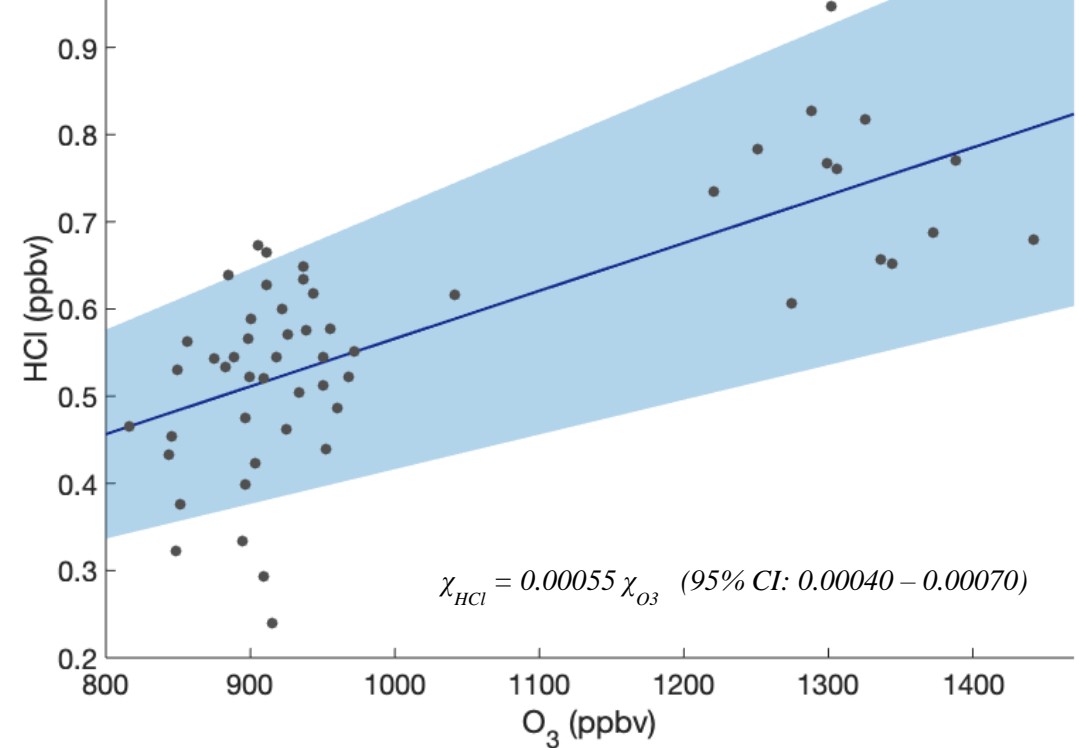

**Figure 17:** Plot of HUSCE $O_3$ v. HCl observations between 65-80 hPa (grey circles). The line corresponds to the slope calculated from the HUSCE observations, and the shaded area represents the 95 % confidence interval range for the regression coefficient.
