# Peer review of "situ observations of stratospheric HCl using three-mirror integrated cavity output spectroscopy"

_Atmospheric Measurement Techniques, 2021_

## Author Comment (AC1)

**Response to Referee 2**

- 1. It would be useful for the reader if the dimensions, weight and power requirements of the RIM-ICOS was provided, as it's deployment on a balloon suggests it could also be applicable to other platforms with payload constraints.
  - 1.1 Response to Referee: Thank you for this suggestion. We've updated the manuscript accordingly.
  - 1.2 Changes to Manuscript: On page 9/lines 280-282, the following text has been added: "The dimensions of the pressure vessel are 1.2 m in length and 0.3 m in diameter. The two pumps were housed in a separate box with dimensions of 27 cm x 41 cm x 16 cm. The HCl instrument platform requires 590 W when operating and weighs 68 kg."
- 2. The authors state that the instrument inlet and measurement cell are held at a temperature of 310 K in order to reduce temperature-related changes to the HCl spectra being fitted and also minimise surface effects from both HCl and water. One possible consequence of this heating is that sampled HCl in the particle or ice phase could repartition into the gas phase under these warmer temperatures. This potential interference has been discussed previously for both stratospheric and tropospheric HCl observations (e.g., Webster et al. 1994 & Crisp et al. 2014). The authors should discuss the potential impact of this repartitioning of HCl on their measurements.
  - 2.1 Response to Referee: Thank you for this interesting point. First, we'll unpack the issues brought up in Webster et al., 1994. They reported on in situ stratospheric measurements in the Arctic, which is more related to our campaign environment than the measurements made in Crisp et al., 2014. Webster et al. report on measurements of HCl, N2O, CH4, NO2, and HNO3. They discuss thermal re-partitioning of HNO3 when aerosols vaporize upon being heated by the instrument. The concern is that the instrument's gas phase HNO3 measurement may be off because some of the HNO3 in the aerosols will enter gas phase as the aerosols vaporize. They discuss only HNO3 in this context because stratospheric aerosols associated with polar stratospheric clouds are commonly nucleated by HNO3, meaning its concentration in a typical aerosol in this environment will be significantly higher than other gas-phase stratospheric species. Even so, Webster finds the error would amount to 1% for 1-micron-diameter particles. Outside the Arctic, stratospheric aerosols are also often nucleated by H2SO4. HCl, however, is not a significant stratospheric nucleation source. Furthermore, Webster does not discuss any errors in their stratospheric HCl measurements arising from thermal re-partitioning or from any other issue related to heating the instrument.

Crisp et al. does explicitly discuss error associated with increasing gas-phase HCl from vaporizing aerosols from heating. Their measurements entail heating the inlet to 308 K, around the same temperature to which our instrument was heated during flight. However, they are measuring tropospheric air near the ocean. The environment they measure therefore has notably different aerosols than what is encountered in the stratosphere—specifically ones formed from sea spray. As such, their observed aerosols are rich in Cl- from the sea salt. Crisp et al. are specifically trying to "assess the impact of HCl volatilization from *chloride-containing* particles" (Crisp et al., Section 2.2). The potential for error in these tropospheric HCl measurements is analogous to the HNO3 measurements in the stratosphere discussed in Webster et al., 1994. Again, stratospheric aerosols are not known to be rich in chloride.

This comment by Referee 2 is certainly important to consider when making in situ atmospheric measurements, and we thank them for bringing it up. However, the referee's comment applies more to  $H_2SO_4$  or  $HNO_3$  in the stratosphere or HCl in the troposphere, especially near bodies of saline water. We do not feel it applies to stratospheric observations of HCl made by our instrument.

2.2 Changes to Manuscript: None

---

## Author Comment (AC2)

**Response to Referee 1:**

- 1. Page 1, Line 13. The abbreviation of MLS is incorrectly marked here.
  - 1.1 Response to Referee: The parenthetical MLS was intended to specify which satellite measurements were compared with our observations. After reading the reviewer's comment, we agree that could be clearer. We've updated the text accordingly.
  - 1.2 Changes to Manuscript: On page 1/lines 13-14, the text is now, "The observations agreed with nearby satellite measurements made by the Earth Observing System Microwave Limb Sounder within 10 % on average."
- 2. *Page 3, Line 67. The response time is related to the volume of the cell and sample flow rate. This argument needs the support of the detailed information of these parameters.*
  - 2.1 Response to Referee: For our instrument, we do state the dimensions of our cell, the sample flow rate of the pumps, and the net flush rate for the cell under normal operation (Section 2.2). The commercial HCl ICOS spectrometers sold by Tiger Optics and Los Gatos Research do not have detailed descriptions published in open literature. Hagen et al., 2014 does report that their CRDS instrument operates at 8 sLpm and that the cavity is 90 cm. However, they do not provide the diameter of their mirrors (and their instrument schematic suggests the cell is not a simple cylinder anyway; Hagen et al., 2014, Figure 3). As such, the volume of their cell is unclear. Therefore, we are unable to provide most of these parameters for the other instruments not developed by us.

What they all do provide is the response time, though, which allows for the most direct comparison with our instrument. The response time for an analyte like HCl is more complicated than the flush rate, which could be determined by the cell volume and sample flow rate. Unlike instruments measuring gases like CO2, the response time of an HCl instrument also depends on factors such as heating of the cell and the material that the cell is composed of. This is because HCl is especially prone to being scavenged by surfaces. As such, the response time is the culmination of a large portfolio of factors beyond cell volume and sample flow rate—many of which are not explicitly provided for the other cited instruments. As such, the simplest and most direct comparison is the one we provide.

- 2.2 Changes to Manuscript: None
- 3. Section 2.2, page 5. What is the bandwidth of the detector? The mirror reflectivity or effective cavity length was determined by the ring-down measurement. What is the ring-down time of the empty cavity? By using re-injection performance, more light will enter the cavity. It is no longer appropriate to use the base length divided by 1-R to express the effective optical path, which is usually used for laser beam one-time injection into the cavity.
  - 3.1 Response to Referee: The bandwidth of the detector and pre-amp used during flight is 1 MHz. The Stirling-cooled detector and pre-amp has a bandwidth of 1.4 MHz. The manuscript has been revised to include this information.

The ring-down time, which we called the cavity time constant, is 7.9 microseconds for an empty cavity. We have added that number to the manuscript and clarified that we are referring to the ring-down time.

The average, effective optical path is still dictated by the base length divided by 1-R as long as the analyte of interest is not present outside the cavity (if it is present in the extra-cavity volume, then some of the light will be absorbed as it reflects between the first cavity mirror and the re-injection mirror, RIM). More light entering the cavity after first being reflected by the RIM does not affect that additional light's average lifetime in the cell once it enters the cell. That is strictly dictated by the length of the cell and the reflectivity of the mirrors.

Additionally, the effective cavity time constant would not be affected the presence of the RIM, as the time taken for the light to reflect off the RIM is orders of magnitude shorter than the amount of time the light spends in the optical cavity.

3.2 Changes to Manuscript: On page 5/line 155-157, the text is now, "During the HUSCE balloon flight, the detector used was a four-stage thermoelectrically cooled MCT detector that, coupled with a pre-amplifier, had a bandwidth of 1 MHz (Vigo, PVI-4TEMXPXX-F)."

On page 8/line 248-249, the text is now, "A two-stage pre-amplifier and anti-aliasing filter collectively adjusts the gain to 5 x  $10^5$  V A-1, with the detector and pre-amp resulting in a bandwidth of 1.4 MHz."

On page 5/lines 144-146, the text is now, "The mirrors have a light loss of 200 ppm at  $3.34 \mu m$  (R = 0.9998, or 99.98 % reflective), which is determined by pulsing light into the cell and measuring the e-folding time for decrease in light intensity (the ring-down time, which is 7.9 microseconds for an empty cavity)."

- 4. Section 4. What kind of interference does "balloon interference" mean? What is the ascent and descent speed of the balloon? Will the release of the helium affect the measurement of HCl? Is the pressure of the sample cell kept constant or the same as the ambient pressure? Will the residence time of the sample in the cavity change?
  - 4.1 Response to Referees: Balloon interference refers to observed air having physically interacted with the surface of the balloon. This is common in balloon-borne campaigns. When air interacts with the balloon surface, the air usually gets 1) some water vapor that outgasses off the balloon, and 2) some thermal energy from the balloon due to its surface being heated by solar radiation. Evidence of balloon interference, therefore, manifests as elevated and highly variably water vapor and temperature levels (stated on page 12/lines 360-362 in the manuscript). This is discussed in more detail in the citation provided in the manuscript: Kräuchi et al., 2016. We've modified the manuscript to better clarify the purpose of this citation. The ascent rate was approximately  $3 \text{ m s}^{-1}$  on average. The descent rate was approximately 2 m s-1 on average and never rose above 5 m s-1. We agree the descent rate should be in the manuscript and have revised accordingly. The release of helium would not affect the measurements of HCl since they were made on descent, and the gondola was approximately 50 meters below the balloon. Also, the release of helium was gradual and sporadic. We have updated the manuscript to better convey the location of the gondola relative to the balloon. The pressure of the cell is kept constant at 53 hPa when feasible—at atmospheric pressure = 60 hPa and above (stated on page 10/lines 309-310).

The pumps are constant volume displacement, so residence time is not significantly affected by pressure changes. The reporting time of 30 seconds for our data would certainly extend beyond any slight change that may occur in residence time.

4.2 Changes to Manuscript: On page 12/line 360-362, the text is now, "There is evidence that balloon interference may have impacted portions of the mid-stratospheric descent, based on anomalous readings from the diagnostic water vapor measurement and the ambient temperature measurement (for more detailed discussion of balloon interference, see Kräuchi et al., 2016)."

On page 10/line 299, the text is now, "The descent rate of the balloon was adjusted in real time, averaging 2 m s-1 and never rising above 5 m s-1."

On page 9/line 279-281, the text is now, "The HCl instrument was secured within a sealed cylindrical pressure vessel and mounted to a gondola suspended approximately 50 meters below the balloon, to separate the instrument platform from the wake of the balloon."

- 5. During the flight, will the changes in the atmospheric temperature affect the performance of the cavity?
  - 5.1 Response to Referee: The cell is temperature-controlled and sealed in a temperaturecontrolled pressure vessel that maintained a constant pressure, so the instrument is isolated from atmospheric temperature variability (page 9/line 281-283 of the manuscript). For that reason, atmospheric temperature changes do not perceptibly affect the performance of the cavity.
  - 5.2 Changes to Manuscript: None

---

## Author Comment (AC3)

Response to L.E.C. Christensen:

1. *Abstract should be clearer: 26 pptv was demonstrated in lab not flight.*
   1.1 Response to Referee: We'll clarify that 26 pptv refers to a laboratory assessment, not the precision in-flight.
   1.2 Changes to Manuscript: On page 1/Lines 11-12, the text is now, "Laboratory assessments demonstrated that the spectrometer has a 90 % response time of 10 s to changes in HCl and a 30 s precision of 26 pptv."

2. *Line 92: Add $O_3$ to list of species*
   2.1 Response to Referee: We'll update this as well.
   2.2 Changes to Manuscript: On page 3/lines 91-92, the text is now, "When selecting the line, the spectral interference from several common stratospheric molecules was considered, including: $CH_4$, CO, $CO_2$, $H_2O$, $O_3$, NO, and $N_2O$."

3. *Line 101: Add instantaneous linewidth and SMSR of ICL laser*
   3.1 Response to Referee: The SMSR of the ICL has been added to the manuscript. The FTIR spectrometers used to characterize the laser don't have a sufficient resolution to measure the instantaneous linewidth (both of them had a resolution of 0.125 $cm^{-1}$). However, we see no error associated with linewidth during measurements at such low pressure that Doppler broadening dominates. We find no measurable difference between the calculated and observed Doppler width, so we believe the instantaneous linewidth to be 0.001 $cm^{-1}$ at most. Because we did not rigorously measure this, though, we do not present this metric in the manuscript.
   3.2 Changes to Manuscript: On page 4/lines 101-102, the text is now, "The laser emission is centered at 2963 $cm^{-1}$, or 3.37 μm, and has a side-mode suppression ratio greater than 25 dB (Borgentun et al., 2015)."

4. *Line 114: You sure this is heating? The main lasing mode can get pulled red from feedback which looks like heating.*
   4.1 Response to Referee: We're providing a general description of the problem of laser feedback. We did not experience these issues for the HCl instrument, as we started with the optical isolator to avoid such feedback. We'll update the manuscript to be clearer about this.
   4.2 Changes to Manuscript: On page 4/lines 113-114, the text is now, "The polarizer and quarter-wave plate (shown collectively as (2) in Fig. 3) serve as an optical isolator that prevents laser light from ultimately reflecting back into the laser housing, which is known to cause unwanted feedback."

5. *Line 161: What pressure did you regulate the cell to during lab experiments – those expected during balloon flight? Later in the manuscript, you mention no regulation during flight experiment below 60 mbar.*
   5.1 Response to Referee: This is a good point, thank you. Unless otherwise stated, the cell pressure in lab was regulated to 53 hPa—the same as during the balloon flight, when feasible.

5.2 Changes to Manuscript: On page 6/line 164-165, the following sentence has been added. "Unless otherwise specified, cell pressure in the laboratory was regulated to 53 hPa."

6. *Do you have any laser spontaneous emission issues?*
6.1 Response to Referee: After receiving this review, we evaluated whether we experience this issue. There are very weak methane absorption lines in our laser scanning region. We created theoretical spectra that showed we could saturate several of these lines if we flowed pure methane through the cell. We did this and found that 4 % of the light was not absorbed at these wavelengths. While the effect is fortunately minor, we very much thank the reviewer for bringing this up. We have re-fit our flight data for HCl, adjusting the calculated laser power impinging on the detector to remove 4 % of the light that is from spontaneous laser emissions. Figures 10 and 17 have been updated with this new analysis. Some of the discussion in the main text that quantifies these comparisons was also modestly affected by this correction and has been updated as well. We have also increased the uncertainty in our accuracy by 1 % to account for the uncertainty in the percentage of measured light that is from spontaneous emissions. The general conclusions regarding the instrument validation via MLS and $O_3$/HCl correlation remain the same. We plan to add a bandpass filter to the optical path to block these spontaneous emissions in future campaigns that involve this instrument. Thank you again for bringing this to our attention.
6.2 Changes to Manuscript: On page 9/lines 273-281, the text is now, "Separately, we evaluated to what extent the laser was emitting light through amplified, spontaneous emission (ASE), incoherent light emitted by the laser at a broad array of wavelengths. While this light from ASE should be orders of magnitude lower than light at the lasing frequency, light that is at wavelengths outside the cavity mirror reflective coating will pass through the cavity and create an apparent offset in the total laser power. To determine whether ASE was an issue in the instrument, very weak methane lines in the scanning region were saturated with pure methane and showed that 4 % of the light is from ASE, which is corrected by decreasing the calculated laser signal by 4 %. This approximately translates to a 4 % increase in HCl mixing ratios that were determined without this correction. The uncertainty in accuracy is 8 % (5 % from uncertainty in spectral parameters; 2 % from uncertainty in ICOS mirror reflectivity and HCl-surface interactions; 1 % from uncertainty in ASE signal). In-flight accuracy is given a conservative upper bound (11 %) due to less stable in-flight conditions."
On page 11/lines 340-342, the text is now, "The spectrum in Fig. 9 (bottom panel) is a 30 s averaged spectrum that corresponds to an ambient mixing ratio of 1.19 ppbv HCl, measured at atmospheric pressure of 26.3 hPa." (Fig. 9 caption was similarly updated)
On page 11/lines 352-353, the text is now, "All observations were greater than *3 x dl*, where *dl* is the detection limit, defined as the noise-equivalent absorption for a 30 second average (70 pptv)."
On page 14/lines 451-452, the text is now, "HUSCE HCl and $O_3$ measurements in the lower stratosphere (65-80 hPa) yield a slope of $0.00057 \pm 0.00007$ (Figure 17)."
In Table 1, reported uncertainty in accuracy for our instrument has been increased by 1 % for both columns.

Figure 10 has been remade with corrected flight data and corrected percent differences compared to average MLS observations (though the mean absolute percent difference is still 8 %).

Figure 17 has been remade with the corrected data, updated regression coefficient (rising from 0.00055 to 0.00057), and updated 95 % CI for that coefficient.

7. *Line 285: Did pre- and post- ringdown times agree? Why isn't ringdown time measured periodically during flight – is it needed?*

   7.1 Response to Referee: While we collected ringdown spectra at the beginning of the flight for the wavenumber affiliated with the HCl absorption feature, every spectrum in flight includes an off period at the end that can serve as a ringdown spectrum. A ringdown measurement is therefore performed with every scan, and they indicate the cavity time constant was consistent throughout the flight. We agree this can be clarified in the manuscript and have updated it accordingly.

   7.2 Changes to Manuscript: On page 10/lines 303-305, the text is now, "The instrument was powered on for 26 minutes prior to recording spectra to 1) allow time for system controls to stabilize, and 2) record ringdown spectra to determine the reflectivity of the ICOS mirrors for the wavelength at which HCl absorbs, R = 0.9998. The measured reflectivity in flight agrees with the measurements made in the laboratory. Ringdowns were also recorded at the end of every spectrum, when the laser is turned off, that was collected during the balloon descent and showed no significant change in the mirror reflectivity throughout the campaign."

8. *Line 320: I'm not following the lineshape in Figure 9. Figure shows water is to blue of HCl while the tail in Figure 9 is red. Is this due to instrument function?*

   8.1 Response to Referee: The theoretical spectrum provided in Figure 2 shows $H_2O^{18}$ absorption at 200 ppmv. This is to illustrate that even extremely elevated $H_2O$ levels only subtly interfere with HCl. However, the $H_2O^{18}$ absorption line blue of the HCl line is not perceptible during the flight. $H_2O$ occasionally--and transiently--reaches up to 150 ppmv, but it is usually closer to 20 ppmv or lower. We did not detect significant water vapor levels when the spectrum in Figure 9 was collected. The red tail on the absorption feature is not $H_2O^{18}$ but rather an instrument function idiosyncratic to ICOS. The red tail is due to the cavity time constant being comparable to the tuning rate of the laser (i.e. the intensity of light entering the cavity at any given time is still decreasing exponentially as the laser scans through the subsequent wavelengths, resulting in a red skew in absorption features). This is taken into account when fitting. Thank you for pointing this out; we will update the description to clarify.

   8.2 Changes to Manuscript: On page 11/lines 343-344, the text is now, "The absorption feature has an asymmetric skew due to the cavity time constant being comparable to the tuning rate of the laser. This skew, characteristic of all ICOS spectra, is taken into account when fitting the HCl absorption feature."

9. *Line 339: Did you have an inlet tube during flight? Did you also Silco coat it? Where was the opening placed – e.g. below the gondola?*

   9.1 Response to Referee: Yes, we did. We'll add a brief description of the in-flight inlet in the manuscript that answers these questions.

9.2 Changes to Manuscript: On pages 9-10/lines 293-296, the following text was added. "The instrument inlet during flight protruded from the center of the pressure vessel, parallel to the length of the vessel. The inlet was a SilcoNert2000 coated steel tube with an inner diameter of 1.1 cm and a length of 62 cm, with the tip being 54 cm away from the side of the gondola. This length was chosen to ensure sampled air was sufficiently separated from the gondola. The inlet was also heated to 310 K and insulated."

*Note that referenced line numbers may differ slightly from the original manuscript due to added text in response to comments.*